# Map-Based Functional Analysis of the *GhNLP* Genes Reveals Their Roles in Enhancing Tolerance to N-Deficiency in Cotton

**DOI:** 10.3390/ijms20194953

**Published:** 2019-10-08

**Authors:** Richard Odongo Magwanga, Joy Nyangasi Kirungu, Pu Lu, Xiaoyan Cai, Zhongli Zhou, Yanchao Xu, Yuqing Hou, Stephen Gaya Agong, Kunbo Wang, Fang Liu

**Affiliations:** 1Research Base in Anyang Institute of Technology, State Key Laboratory of Cotton Biology/Institute of Cotton Research, Chinese Academy of Agricultural Science (ICR, CAAS), Anyang 455000, China; magwangarichard@yahoo.com (R.O.M.); nyangasijoy@yahoo.com (J.N.K.); lupu1992@cricaas.com.cn (P.L.); caixy@cricaas.com.cn (X.C.); zhouzl@cricaas.com.cn (Z.Z.); xuyanchao2016@163.com (Y.X.); houyp@cricaas.com.cn (Y.H.); 2School of Biological and Physical sciences (SBPS), Main campus, Jaramogi Oginga Odinga University of Science and Technology (JOOUST), P.O. Box 210-40601, Bondo, Kenya; sgagong@jooust.ac.ke

**Keywords:** nitrogen fertilizer, nodule-inception-like proteins, cotton plant, miRNAs, oxidant and antioxidant enzymes, virus-induced gene silencing (VIGS) plants

## Abstract

Nitrogen is a key macronutrient needed by plants to boost their production, but the development of cotton genotypes through conventional approaches has hit a bottleneck due to the narrow genetic base of the elite cotton cultivars, due to intensive selection and inbreeding. Based on our previous research, in which the BC_2_F_2_ generations developed from two upland cotton genotypes, an abiotic stress-tolerant genotype, *G. tomentosum* (donor parent) and a highly-susceptible, and a highly-susceptible, but very productive, *G. hirsutum* (recurrent parent), were profiled under drought stress conditions. The phenotypic and the genotypic data generated through genotyping by sequencing (GBS) were integrated to map drought-tolerant quantitative trait loci (QTLs). Within the stable QTLs region for the various drought tolerance traits, a nodule-inception-like protein (NLP) gene was identified. We performed a phylogenetic analysis of the NLP proteins, mapped their chromosomal positions, intron-exon structures and conducted ds/dn analysis, which showed that most *NLP* genes underwent negative or purifying selection. Moreover, the functions of one of the highly upregulated genes, *Gh_A05G3286 (Gh NLP5)*, were evaluated using the virus gene silencing (VIGS) mechanism. A total of 226 proteins encoded by the *NLP* genes were identified, with 105, 61, and 60 in *Gossypium hirsutum, G. raimondii*, and *G. arboreum*, respectively. Comprehensive Insilico analysis revealed that the proteins encoded by the *NLP* genes had varying molecular weights, protein lengths, isoelectric points (*pI*), and grand hydropathy values (GRAVY). The GRAVY values ranged from a negative one to zero, showing that proteins were hydrophilic. Moreover, various *cis*-regulatory elements that are the binding sites for stress-associated transcription factors were found in the promoters of various *NLP* genes. In addition, many miRNAs were predicted to target *NLP* genes, notably miR167a, miR167b, miR160, and miR167 that were previously shown to target five *NAC* genes, including *NAC1* and *CUC1,* under N-limited conditions. The real-time quantitative polymerase chain reaction (RT-qPCR) analysis, revealed that five genes, *Gh_D02G2018, Gh_A12G0439, Gh_A03G0493, Gh_A03G1178*, and *Gh_A05G3286* were significantly upregulated and perhaps could be the key *NLP* genes regulating plant response under N-limited conditions. Furthermore, the knockdown of the *Gh_A05G3286 (GhNLP5)* gene by virus-induced silencing (VIGS) significantly reduced the ability of these plants to the knockdown of the *Gh_A05G3286* (GhNLP5) gene by virus-induced gene silencing (VIGS) significantly reduced the ability of the VIGS-plants to tolerate N-limited conditions compared to the wild types (WT). The VIGS-plants registered lower chlorophyll content, fresh shoot biomass, and fresh root biomass, addition to higher levels of malondialdehyde (MDA) and significantly reduced levels of proline, and superoxide dismutase (SOD) compared to the WT under N-limited conditions. Subsequently, the expression levels of the Nitrogen-stress responsive genes, *GhTap46, GhRPL18A*, and *GhKLU* were shown to be significantly downregulated in VIGS-plants compared to their WT under N-limited conditions. The downregulation of the nitrogen-stress responsive genes provided evidence that the silenced gene had an integral role in enhancing cotton plant tolerance to N-limited conditions.

## 1. Introduction

Nitrogen (N) is an essential macro-element for all forms of life [1]. Plants and fungi are the only eukaryotic organisms, which are able to assimilate inorganic N. Among the higher plants, it is only the leguminous plants that are the most common or the most important group of plants that can form associations with N-fixing bacteria, known as the *Rhizobiae*, which enables them to fix free atmospheric nitrogen [2]. The bacteria colonize the roots, forming nodules, the nodule formation is governed by a protein known as nodule inception-like protein (NLP) [3]. The main source of nitrogen for non-leguminous plants, is through nitrogen fertilizer application, however, boosting soil fertility through nitrogen application is expensive and requires constant soil moisture levels [4]. Therefore, farmers and breeders have employed several strategies to solve the problem of N-deficiency through the initiation of conventional approaches, such as crop rotation and intercropping with leguminous plants, though minimal success has been achieved [5]. The only mechanism to boost cotton production under an N-limited environment is by adopting molecular approaches, by determining the suitable plant transcription factors with a higher role in enhancing the nitrogen metabolism in plants. Some of the genes known to be highly correlated to nitrogen and nitrogen metabolism in plants are the members of the nodule-inception-like protein (NLP) family, they are known to be highly conserved and enhance plant’s response to N-deficiency [6]. Plants being sessile, they are constantly exposed to various environmental stresses, including growing in nitrogen (N) deficient conditions. The plants have evolved various survival strategies to tolerate the low N levels in the soil one of which is through the induction of the NLP proteins in order to enhance their survival under such conditions. Plant response to N-deficiency usually begins with limitations in uptake. Nitrogen exists in the soil as nitrate nitrogen, ammonium nitrogen, amino acids, proteins, and other nitrogenous substances, the cotton plant being non-leguminous, can only use inorganic forms of N, either as nitrate (NO_3_^−^) or ammonium (NH_4_^+^) [7]. Nitrate is the principal source of N since ammonium is quickly transformed to nitrate in the soil solution, just like other higher plants, cotton plants absorb nitrate through the roots and transport it directly to the leaves through the transpiration stream. Once on the leaf, nitrate is reduced to ammonium and combined with organic acids to form amino acids and proteins [8].

The nodule inception-like proteins (NLPs) are widely distributed across the plant species, even to non-leguminous plants [9]. Even in non-leguminous plants, they function as nodule initiators and regulate the number of nodules that are formed [2]. For instance, in maize, a total of 9 *ZmNLPs* have been identified and found to have an integral role in the primary response to nitrogen [10]. Moreover, overexpression of maize NODULE-INCEPTION-like proteins, *ZmNLP6*, and *ZmNLP8* in the Arabidopsis, restored nitrate assimilation and induction of nitrate-responsive genes in the *NLP7-4* mutant Arabidopsis [11]. The restoration of nitrate assimilation in the mutant Arabidopsis showed that the maize *NLP* genes had a similar regulatory role to the Arabidopsis *NLP7* gene (*AtNLP7*) in nitrate signaling and metabolism [11]. Moreover, studies have shown that plants have developed an advanced mechanism to deal with nitrogen deficiency, through uptake, transport, and assimilation of nitrogen under N-limited environment [11]. Furthermore, the plants only utilize 30% of the applied nitrogenous fertilizers, while the rest is either lost through leaching and or fixed in the soil [12].

The nitrate form of nitrogen is mainly translocated in the plants through two families of Nitrate transporters (NRT/NPF), the nitrate transporter 1/peptide transporter family previously known as the *NRT1/PTR* family and the *NRT2* [13]. In Arabidopsis, there are 7 and 53 members in the *NRT2* and *NPF* families, respectively [14]. The phosphorylation of the *NRT1.1* by two kinds of calcineurin B-like (CBL)-interacting protein kinases, *CIPK8* and *CIPK23* enhance the activity of the NRT1.1 in plants by improving their response to either N-deficient or higher N concentration level conditions [15]. Moreover, *NRT1.1* has dual-affinity of N transportation and or nitrate-sensing functions, acting in the upstream region of the Arabidopsis nitrate regulated-1 (ANR1), in the N signaling pathway [16]. The members of the *NRT2* transporters have high-affinity to nitrate, and the majority of them require *NAR2* (NRT3), to facilitate the transportation of nitrate within the plants [17]. Several studies have shown that the high nitrate affinity transporters do play an integral role in N-uptake efficiency under N-limited environment [18]. Furthermore, analysis of the model plant, *A. thaliana*
*NRT2* genes showed that *AtNRT2.1*, *AtNRT2.2*, *AtNRT2.4*, and *AtNRT2.5* were highly upregulated in the roots compared to other tissues under nitrogen-deficient conditions [19]. Moreover, analysis of the VIGS-plants showed that *AtNRT2.1*, *AtNRT2.2*, *AtNRT2.4,* and *AtNRT2.5*, *NRT2* transporters accounted for over 95% of high-affinity nitrate influx activity under N-limited condition, in which the *AtNRT2.1* was the dominant [20].

Due to cost aspects and environment effects of nitrogen fertilizers, there is increased pressure to develop new crop genotypes with improved performance under N-limited conditions or with minimal N-application [21]. Due to the significance of nitrogen in plants, improving nitrogen uptake efficiency is vital, thus it is important to identify the intrinsic traits plants possess to improve their nitrogen intake efficiency under N-limited environments. Even though a great deal of progress has been made in elucidating the roles of genes and signal pathways that control the N acquisition and plant root phenology in the model plant, but, little progress has been made in understanding the role of N transporters in cotton. in the past, there has been a positive correlation between the quantitative trait loci (QTLs) for nitrogen absorption and plant traits such as the root architecture [22]. Furthermore, other studies have shown that the application of Ammonium chloride and Sodium nitrite increased cell membrane stability (CMS) in wheat plants up-to 46.62% and 84.46%, respectively compared to non-fertilized plants [23]. Moreover, N-deficiency induces cell wall loosening, thereby affecting the cell membrane stability [24].

Cotton is an important industrial crop, the primary source of natural fiber [25]. The commonly grown cotton cultivars are the tetraploid type (4n), which emerged due to polyploidization between two diploid cotton genomes, A and D [26]. The parental line of the tetraploid cotton (AD), are believed to be *Gossypium arboreum* of the A genome and *Gossypium raimondii* of the D genome [27]. Completion of the genome sequences of *G. hirsutum* [28], *G. arboreum* [29], and *G. raimondii* [30] provides valuable resources for exploring the whole cotton genome assembly and their evolution pattern. In the analysis of QTLs in the backcross inbred lines, (BC_2_F_2_) population derived from two tetraploid cotton. The drought-tolerant donor parent *Gossypium tomentosum* and drought-sensitive parental line, *G. hirsutum*, under drought condition, Magwanga et al. (in press), found a stable QTL, contributed by the drought-tolerant parental line, *G. tomentosum*, and determining the genes within the QTL flanking region, *Gh_A05G3286*, a member of the NLP5 Protein was found to be the gene responsible. Therefore, in this research work, we carried out genome wide identification of the NLP proteins in cotton, determined their evolution pattern, physiochemical properties, chromosome mapping, their gene structure, expression level under N-limited condition. Moreover, we further characterized the function of the *Gh_A05G3286* gene through virus-induced gene silencing (VIGS) mechanisms in order to determine the possible role of the protein encoded by the *NLP* genes in cotton under N-limited conditions. The results obtained provide the foundation for the role of the NLP proteins in cotton and thus allowing for future exploration of these genes in developing a high nitrogen use efficient cotton genotypes.

## 2. Results

### 2.1. Identification and Sequence Analysis of the Cotton NLP Proteins

To identify the *NLP* genes in the three cotton sequenced species of A, D, and AD genomes, the conserved domain of NLP protein and the NLP-related domains, RWP-RK (PF02042) and Phox and Bem1 (PBl), with the Pfam number of PF00564 were downloaded from the protein families (PFAM) database (http://pfam.sanger.ac.uk/). The Hidden Markov models (HMM) profile of the NLP protein was subsequently employed as a query to perform an HMMER search (http://hmmer.janelia.org/) against the *G. hirsutum*, *G. raimondii*, and *G. arboreum.* In the identification of the NLP proteins encoded by the *NLP* genes in cotton, two hundred and 26 NLP proteins were identified, with 105, 61, and 60 distribution in *G. hirsutum, G. raimondii*, and *G. arboreum*, respectively. All the two conserved NLP protein domains, RWP-RK (PF02042) and PB1 (PF00564) were identified in all the three cotton species. However, the numbers of NPL proteins of the PB1 (PF00564) domain were higher in proportion compared to those of the RWP-RK domain across the three cotton species. The two domain proportions were in the ratio of one RWP-RK: Three PB1, which indicated that the cotton NLP proteins of the PB1 were the most abundant. The number of NLP proteins identified in cotton was relatively higher compared to the proportions so far identified in other plants such as maize, with nine NLP proteins [10], 56 NLP proteins in the Brassica species [6] and only 9 in Arabidopsis [31]. Moreover, in the evaluation of their physiochemical properties through an online tool, the ExPASy Server (http://www.web.xpasy.org/compute_pi/). The cotton NLP protein characteristics were varied, in which the molecular weights ranged from 7.169 kDa to 151.633 kDa, grand hydropathy values (GRAVY) ranged from −0.904 to a maximum value of 0.399, protein lengths ranged from 63 aa to 1403 aa (Appendix A). The results obtained were consistent with previous findings, such as the analysis of the NLP proteins in Brassica species [6].

### 2.2. Phylogenetic Tree Analysis

To determine the evolutionary pattern of the cotton NLP proteins, the NLP proteins sequences of *G. hirsutum*, *G. raimondii,* and *G. arboreum* were retrieved from cotton functional genome database (https://cottonfgd.org/), while the sequences for *A. thaliana, T. cacao*, and *Vitis vinifera* were retrieved phytozome (https://phytozome.jgi.doe.gov/pz/portal.html). All the proteins sequences were aligned, by adopting multiple sequence alignment using ClustalW a component of MEGA 6 tool. The phylogenetic tree was then generated, by adopting neighboring joint (NJ) and Maximum Likelihood Method with P distance Model, with a 1000 bootstrap replications, site cutoff coverage of 100% and missing gaps/missing data treatment with complete deletion [32]. In the analysis of the proteins encoded by the *NPL* genes in cotton and other plants, the NPL proteins were classified into three main groups, with members of group three being the largest (Figure 1 and Appendix A). In the previous studies of the *NPL* genes in various plants, such as Arabidopsis, the NLP proteins were classified into three clades, designated as *NLP1*, *NLP2,* and *NLP3* [31]. This showed that the cotton NLP protein classification was congruent to previous studies, which could perhaps mean that the NLP proteins are highly conserved. In analyzing the possibility of orthologous gene pair formation between the cotton NLPs and other plants used in the phylogenetic tree, only two genes were found to form orthologous gene pairs with the upland cotton *NLP* genes, *Gh_D05G2521*, and *LOC_Os11g30350* (*NLP* gene from *Oryza sativa*), and the other pair was, *Ghsca101252G01* and *GSVIVG01026649001* (*NLP* gene from *Vitis vinifera*). None of the orthologous gene pairs was formed between cotton and the NLP proteins obtained for *T. cacao* the closest relative of the genus *Gossypium* compared to other plant species analyzed. However, the majority of the cotton NLP proteins were members of group 1 and 2, while few were located in group 3 despite being the largest group. In the analysis of the selected type, we evaluated the synonymous (ds) and non-synonymous (dn) rate of substitution. The ds/dn ratio is significant in evaluating the type of selection pressure, which acted on the proteins encoded by the *NLP* genes, the ds/dn value greater than 1 indicates beneficial selection, those that are less than 1 signify negative selection or purifying selection and those of unit value shows that the mutational effect was neutral [33]. The lowest ds/dn value was estimated at 0.389 for two ortholog genes of *G. raimondii*, while the highest ds/dn value was estimated to be 4.7615 for two paralogous gene pairs between *G. arboreum* and *G. hirsutum* (Appendix A). However, the majority of the gene pairs had their ds/dn values of less than one, which indicated that a number of the NLP paralog gene pairs underwent negative/purifying selection. The result obtained is in agreement with previous investigations on the stress-responsive genes, the late embryogenesis abundant proteins (LEAs), in which the majority of the paralogous gene pairs had da/ds values of less than one [34].

### 2.3. Chromosome Mapping and Subcellular Localization Prediction Analysis of the Cotton Proteins Encoded by the NLP Genes in Cotton

To determine the chromosomal locations of the cotton *NLP* genes based on their positions, data retrieved from the whole cotton genome sequences were used. Chromosome distribution was done using the Basic Local Alignment Search Tool Nucleotide (BLASTN) search against *G. hirsutum* and *G. arboreum* in the cotton genome project (https://jgi.doe.gov/cotton-genome-project-in-al-com/) and *G. raimondii* genome database in Phytozome (http://www.phytozome.net/cotton.php). Out of the 105 proteins found to be encoded by the *NLP* genes in *G. hirsutum*, all were distributed across the 26 chromosomes with only three genes being in the scaffold region. More of the proteins were located in the Dt_sub genome with 53 (50.5%) compared to 49 (46.7%) proteins located in the At_sub genome chromosomes. The highest number of gene loci was noted in chromosome A_h_05 and D_h_05 with eight and nine genes, respectively. While the lowest gene loci densities were observed in A_h_04, A_h_10, and D_h_10 with a single gene in each (Figure 2A). The gene distribution pattern within the two diploid cotton species was similar, all their 13 chromosomes were found to harbor at least one gene. In A genome, the highest gene loci were noted in chromosome A_2_08, A_2_01, A_2_03, and A_2_05 with 6, 7, 7 and 10 genes, respectively, while chromosome A_2_10 harbored only one gene (Figure 2B). In the *G. raimondii* of the D genome, chromosome D_5_03, D_5_04, D_5_05, and D_5_09 with 6, 6, 8, and 9 genes, respectively (Figure 2C). In the evaluation of the subcellular localization of the cotton proteins encoded by the *NLP* genes, an online tool Wolfsport (https://www.Wolfpsort.hgc.jp/) was employed. The coding sequences (CDS) of all the proteins encoded by the *NLP* genes were retrieved from cotton functional genome database (https://cottonfgd.org/analyze/). Then uploaded to an online tool Wolfsport (https://www.Wolfpsort.hgc.jp/) to predict the subcellular localization of the proteins encoded by the cotton *NLP* genes. The tool provides a different multi-location with varying probability, and the site with the highest probability becomes the main site in which the various proteins are likely to be found. The nucleus was the dominant subcellular structure with 51, 33, and 29 proteins encoded by the *NLP* genes in *G. hirsutum, G. raimondii,* and *G. arboreum*, respectively. The number of NLP proteins predicted to be sublocalized within the nucleus accounted for 48.57%, 47.54% and 55% of all the NLP proteins in *G. hirsutum, G. arboreum* and *G. raimondii*, respectively the other subcellular structures observed to harbor the proteins encoded by the cotton *NLP* genes were endoplasmic reticulum (E.R), plasma membrane, mitochondrion, and extracellular structures. In *G. hirsutum*, 26 (24.76%) proteins were found to be localized within the E.R, 24 (22.86%) in the plasma membrane, 3 (2.86%) in mitochondria and only one (0.95%) in the extracellular structures. In the two diploid cotton species, *G. raimondii*, 14 (23.33%) proteins were predicted to be located within the E.R, 12 (20%) in the plasma membrane, 1 (1.67%) located in the cytoplasm and the other single protein was harbored in the extracellular structures. There was minimal variation in *G. arboreum*, 15 (24.59%) in E.R, 15 (24.59%) in the plasma membrane and a single protein accounting for 1.64% of all the NLP proteins in *G. arboreum* was predicted to be embedded within the cytoplasm.

### 2.4. Gene Structure Analysis of the Cotton NLP Proteins

To gain further information into the structural diversity of cotton *NLP* genes, the exon/intron organization in the full-length cDNAs with their corresponding genomic DNA sequences of individual *NLP* genes in cotton were analyzed using an online, the gene structure displayer server (http://gsds.cbi.pku.edu.cn/). In the analysis of the gene structures of the upland cotton, *G. hirsutum* almost all the genes were disrupted by intron except 13 genes, such as *Gh_Sca101252G01* (uncharacterized), *Gh_D01G0769* (tetratricopeptide repeat protein 1), *Gh_A01G0750* (tetratricopeptide repeat protein 1), *Gh_D13G1358* (tetratricopeptide repeat protein 1), *Gh_A13G1093* (tetratricopeptide repeat protein 1), *Gh_D04G1546* (tetratricopeptide repeat protein 1), *Gh_D04G0995* (uncharacterized), *Gh_D02G2018* (serine/threonine-protein phosphatase 4 regulatory subunit 3), *Gh_A03G1567* (uncharacterized), *Gh_A05G2263* (uncharacterized), *Gh_D05G2522* (uncharacterized), *Gh_D05G2083* (uncharacterized), and *Gh_D05G3757* (uncharacterized). The level of intron disruption ranged from a single intron to a maximum of thirteen introns as evident among the following genes *Gh_A03G0454*, *Gh_A07G0445*, *Gh_A11G3016*, *Gh_D03G1084*, *Gh_D07G0509*, *Gh_D09G0055*, *Gh_D11G0397,* and *Gh_D11G0626* (Figure 3A). In relation to the two diploid cotton species, in *G. arboreum*, the cotton species of the A genome, only five genes were found to be intronless while the rest were interrupted either one to a maximum of thirteen introns. The highest level of intron disruption was observed in five genes, *Ga01G2121*, *Ga07G0597*, *Ga11G3460*, *Ga09G0065,* and *Ga11G3698* with 13 introns *(*Figure 3B). Moreover, the same trend was also noted in *G. raimondii*, the cotton species of the D genome, in which the member of the CBS domain-containing protein CBSCBSPB5 was the most interrupted by introns (Figure 3C). Previous investigations have shown that overexpression of the CBS-domain containing protein in rice improved tolerance levels of transgenic tobacco to oxidative stress, salinity and heavy metal toxicity [35]. Moreover, expression of a CBS domain-containing protein, *GmCBS21* a candidate gene for nitrogen use efficiency (NUE) was found to enhance abiotic stress tolerance and improved performance of transgenic *Arabidopsis thaliana* under low nitrogen condition [36]. However, the exact role of the intron requires further investigation, though we preempt that the presence of introns may not be causing any alteration on the gene action.

### 2.5. Cis-regulatory Element Analysis and miRNA Target Prediction on the Cotton NLP Genes

To determine the *cis-*regulatory element, a 1 kb up and down stream region of the sequence of each gene was submitted to an online tool the Plant CARE (http://bioinformatics.psb.ugent.be/webtools/plantcare/html/) to obtain the various *cis-*regulatory elements associated with the *NLP* genes. The *cis-*regulatory sequences are linear nucleotide fragments of non-coding DNA with the main role of regulating gene expression and in turn, controls the development and physiology of an organism [37]. We sought to determine any possible interactions of the various cotton *NLP* genes to any of the known *cis-*regulatory elements. We found that a number of the genes were associated with various elements such as MYCCONSENSUSAT (CANNTG), which functions as an elicitor in abiotic stress signaling in plants, MYB2AT (TAACTG) mainly induced by dehydration stress, MYBCORE (CNGTTR) a *cis-*regulatory element that is responsive to water stress among others (Appendix A). The detection of these myriads of *cis-*regulatory elements showed that these genes are critical in enhancing the plant’s survival under abiotic stress conditions. Similar *cis-*regulatory elements have been found to correlate to various plant transcriptome factors such as the *NAC*, *MYB*, *LEA* genes among others [38]. Moreover, the miRNA targeting the various cotton *NLP* genes were predicted. The CDS of all the cotton NLPs were retrieved from cotton functional genome database (https://cottonfgd.org/analyze/). The retrieved sequences were uploaded onto the online tool the psRNATarget server (http://plantgrn.noble.org/psRNATarget/), non-cotton miRNAs were removed and only cotton miRNAs were analysed A microRNA (miRNA) is a small non-coding RNA molecule with approximately 21 nucleotides in length, found in plants, animals, and some viruses mainly function in transcriptional and post-transcriptional regulation of gene expression [39]. We sought to investigate any possible target by the miRNAs, for the *NLP* genes obtained from the tetraploid cotton 67 genes were found to be targeted by 47 miRNAs. Four (4) sets of miRNAs were found to have the highest level of genes associated with them, such as ghr-miR7484a (14 genes), ghr-miR7484b (14 genes), ghr-miR7495a (12 genes) and ghr-miR7495b (12 genes). previous studies in cotton have shown that miR7484a has a functional role in cotton under drought stress conditions [40]. More interestingly, some of the genes were found to be targeted by very high numbers of miRNAs of 10 and above, *Gh_A07G1460* (10 miRNAs), *Gh_A08G1723* (11 miRNAs), *Gh_D07G1556* (11 miRNAs), and *Gh_D08G2074* (10 miRNAs), while the rested with miRNAs ranging from about one to a maximum of nine miRNAs (Appendix A). No miRNA was found to target any of the *NLP* genes obtained from *G. arboreum*, but a huge number of miRNAs were found to target the *NLP* genes obtained from *G. raimondii*, 171 miRNAs were found to target 57 *NLP* genes (Appendix A). Previous reports have shown that miRNAs play significant roles in enhancing plants survival under extreme conditions. For instance, miR164 has been found to target the *NAC* gene through cleavage, the miR164 direct cleavage to *NAC* gene family in maize, *zmNAC1* enhance lateral root growth in maize and in turn improves their performance under water limiting condition [41]. Furthermore, miR164 targeted four different genes, *Gh_A03G0443*, *Gh_A12G0439*, *Gh_D03G1095*, and *Gh_D12G0440*. Moreover, a miR169a has been found to play the mediation role in N-limited environments, and its overexpression improved transgenic tomato performance under N-limited condition [42]. In our investigation, miR169a targeted *Gh_A08G1723*, *Gh_A09G0059*, *Gh_D08G2074*, and *Gh_D09G0055*, which demonstrated that the NLP proteins could be playing a significant role in enhancing cotton plant’s tolerance to N-limited conditions.

### 2.6. RT-qPCR Validation of the Selected GhNLP Genes

Based on the phylogenetic tree and gene structures of the upland cotton, *NLP* genes. Fifty three genes were used for determining their expression levels in the tissues of *G. hirsutum* exposed to a nitrogen-limited condition. The genes were classified into three groups. Group 1 members were highly upregulated in all the tissues examined. The group 2 members, exhibited partial upregulation, which implied that they exhibited normal expression range while the group three members, showed differential expression, across the three tissues examined and at different time points (Figure 4). The RT-qPCR analysis was done on the three plant organs, roots, stems, and leaves. The plants were exposed to N-limited conditions, and tissues collected at 0 h, 3 h, 6 h, and 12 h of post-stress exposure. Cotton *GhActin* was used as the reference gene. Based on the RT-qPCR analysis, five (5 genes) were found to exhibit significantly higher expression levels, *Gh_D02G2018, Gh_A12G0439, Gh_A03G0493, Gh_A03G1178*, and *Gh_A05G3286,* perhaps could be the candidate genes responsible for enhancing cotton plants to tolerate low nitrogen level environments. Moreover, *Gh_A05G3286 (NLP5)* was further evaluated, being it showed significantly higher upregulation under N-limited conditions among all the five genes.

### 2.7. Silencing of Gh_A05G3286 (NLP5), Physiological and Morphological Evaluation of the VIGS-Plants and the Non-Cotton Seedlings Under Nitrogen Limited Condition

In order to understand the role of the NLP proteins encoded by the upland cotton NLP genes, a highly upregulated gene as determined by the RNA seq and RT-qPCR validation, *Gh_A05G3286 (NLP5)*. The *NLP* gene, *Gh_A05G3286 (NLP5)* with a sequence length of 819 bp fragment was amplified from cDNA using the VIGS-F (5′ACACGTGCTTGGACTCTGTC3′) and VIGS-R (5′CGAATTTGATGTCAGCGCGT3′) primers. The fragment was amplified using KAPA HiFi HotStart ReadyMix, cloned into the pTRV2 vector to yield pTRV2:Gh_A05G3286 (NLP5) and verified through sequencing. The construction of the pTRV2 silencing vector and method of viral inoculation followed the protocol by Scofield et al. [43]. The phytoene desaturase (PDS) was used in order to determine the effectiveness of the vector [38]. After seven days of post infiltration, the plants infused with the vector containing the TRV:PDS exhibited an albino appearance on their leaves, which progressed to 100% bleaching after fourteen days (Figure 5A). The plants infused with the PDS showed albino-like traits on their leaves. The PDS-infused plants behave like wild-type plants in their capacity to etiolate and produce anthocyanins indicating that the light signal transduction pathway seems to be unaffected, the effect of PDS on the plant is similar to CLA1-1 also referred to cloroplastos alterados or altered chloroplast) [44]. In addition, we analyzed the gene-silenced plants and wild types in order to determine the expression levels and the net effect of the knockdown of the gene. The RNA inference is a highly conserved process in eukaryotes, which do, involves the degradation of the target messenger RNAs (mRNAs) using sequence-specific small RNAs, which do result in a reduction level in expression or knockdown of the target gene. The gene knockdown is achieved via small RNA (sRNA) pathways that use both endogenous and exogenous short interfering RNA (siRNA) and microRNA (miRNA) pathways, the pathways do. interact to form a well-balanced gene regulation system [45]. The expression of the target gene was sharply reduced in the leaf tissues of the VIGS-plants but highly upregulated in the leaf tissues of the wild plants, an indication that the expression level of the gene was significantly suppressed (Figure 5B,C). Moreover, the expression level of the target in the VIGS-plants and the wild types under normal condition was significantly reduced two-fold, however, in the wild type and the TRV: 00 plants, the target gene was significantly upregulated compared to the VIGS-plants. The TRV:00 is the empty-vector control, this was done in order to determine the vector had any effect on the plant, the performance of the plants infused with TRV:00 and the wild types exhibited no significant differences [46]. The albino phenotypic appearance and downregulation as revealed by the RT-qPCR results indicated that the role of the knocked gene was significantly suppressed. The results were in agreement with previous findings, which recorded that the expression of a gene is known to be downregulated through RNAi when the albino appearance is more or equal to 75% [47]. Moreover, N-limitation has a negative impact on chlorophyll content, being N deficiency increases the rate of senescence as a result of the rapid decline in chlorophyll content and soluble proteins [48]. Since, Nitrogen is an essential macro-nutrient element of all the amino acids which are the building blocks of plant proteins, which are vital for the growth and development of important tissues and cells like the cell membranes and chlorophyll content. There was a significant reduction in chlorophyll content on the VIGS-plants compared to the wild types under N-limited environment, the reduction in chlorophyll is an indication that the plant’s ability to tolerate low or N-limited conditions was significantly affected thus the decline in chlorophyll content (Figure 5D). Furthermore, we evaluated the fresh shoot, and root biomass of the VIGS-plants and the wild types under N-limited conditions, the VIGS-plants shoot, and root biomasses were significantly reduced compared to their wild types (Figure 5E,F). The results obtained were in agreement with previous findings, which have shown that reduction in N, leads to a reduction in plant growth and development, and in turn, lowers their overall biomass accumulations [49].

### 2.8. Transcripts Investigation of Nitrogen Stress-Responsive Genes, analysis of Oxidant, and Antioxidant Content on the Tissues of VIGS and Non-VIGS Cotton Seedling Exposed to Nitrogen Limited Condition

In order to understand the effect of *GhNLP* gene suppression in the VIGS-plants, we carried out RT-qPCR expression analysis of some of the nitrogen deficiency responsive genes such as 60S ribosomal protein 18A (*RPL18A*), PP2A regulatory subunit (*Tap46*) [50], and cytochrome P450 CYP78A5 monooxygenase (*KLU*) [51]. All the three genes expression was downregulated in the VIGS-plants but was upregulated in the wild cotton species (Figure 6A i–iii). Several approaches have manipulated N-metabolism and transporter genes to increase N use efficiency in plants. For instance, overexpression of cytosolic GS increased plant height and dry weight under low-N conditions in tobacco [52]. Transgenic maize constitutively overexpressing *GLN1-3*, which encodes a cytosolic GS, in leaves, showed a 30% increase in kernel number [53]. The SOD and proline, together with malondialdehyde (MDA) were evaluated. The MDA concentration level was higher in the leaf tissues of the VIGS cotton seedlings, whereas proline and SOD levels were significantly reduced (Figure 6B i–iii). The reduced concentration levels of the antioxidant enzyme assayed revealed that VIGS cotton was highly affected compared to their wild types. Hydrogen peroxide is known as a plant stress signaling compound, and its increased level within the plant tissues triggers the release of various antioxidant enzymes [54]. However, when antioxidants are not induced, the H_2_O_2_ levels becomes lethal, resulting in oxidative damage to the plant cells, and eventually lead to cell death [55]. The excessively produced H_2_O_2_ is down-regulated by various antioxidant enzymes. However, in this research work, we found that the SOD concentration level was significantly low, an indication that the VIGS-plants suffered an extensive oxidative damage, which was further evidenced by the concentration, levels of MDA. Under stress condition, the level of ROS becomes elevated in the plant tissues as a result of uncontrolled process in the electron transport chain and accumulation of photoreducing power. This excess of electrochemical energy can be dissipated through the Mehler reaction, resulting in ROS production, including hydrogen peroxide [56,57], and damage of membranes, reflected in elevated MDA levels.

## 3. Discussion

Enhanced use of nitrogenous fertilizers has led to increased production, change of soil micro and macro-environments, and massive pollution of various water bodies, as a result there is a massive wash off nitrogenous compounds from the agricultural fields leading to eutrophication [58]. For effective application and utilization of nitrogenous fertilizers, the adequate soil moisture content is necessary, thus this requires sufficient rainfall amounts and or maintenance of constant soil moisture through irrigation [59]. The erratic weather pattern, low amount of precipitation and alkalization of arable lands, has affected nitrogen fertilizer application and nitrogen use efficiency, leading to massive losses in agricultural production. The pressure on the arable lands due to human settlement and food demand has led to non-edible crops such as cotton cultivation to be restricted to a much drier, more saline, low nutrient load and heavily polluted lands with heavy metal [60]. Thus in order to maintain cotton production, nitrogen fertilizer application might not be the only solution, being freshwater for agricultural production is also a limiting factor. The plants with the ability to fix free atmospheric nitrogen have been found to have an important protein known as the NODULE-INCEPTION-like proteins (NLPs), this protein has a critical role in plants under N-deficiency condition [9]. In the recent past, the NLPs proteins have also been discovered in non-leguminous plants and found to play an important role in improving the plant’s performance under N-limited environments [61]. The purpose of this experiment was to identify *NLP* genes that were significantly up-regulated under N-deficiency and might therefore be candidates for regulating N-deficiency responses.

In the whole genome identification of the cotton NLP proteins, we found a relatively higher number of the NLP proteins in the three cotton species, the upland cotton *G. hirsutum* harbored almost twice the number of proteins contained in either of the two diploid kinds of cotton combined. A total of 105 NLP proteins were identified in *G. hirsutum*, 60 and 61 in *G. arboreum* and *G. raimondii*, respectively. The overall number of the proteins encoded by the *NLP* genes in *G. hirsutum* was lower than the sum total of the two diploid cotton species, the low number of the NLP proteins in the tetraploid cotton could be attributed to either gene loss or chromosome rearrangement as a consequence of whole-genome duplication. Analysis of various functional genes in the three cotton species tend to have a similar pattern, for instance, in the whole genome identification of the calcineurin B-like (CBL) proteins in the three cotton species, 13, 13, and 22 were identified in *G. raimondii, G. arboreum,* and the cultivated tetraploid cotton, *G. hirsutum* [62]. Similarly, in the identification of the *LEA* genes in cotton, 242, 136 and 142 proteins encoded by the *LEA* genes were identified in *G. hirsutum*, *G. arboreum,* and *G. raimondii*, respectively [34]. The number of NLP proteins in cotton, seems to be significantly higher than the number obtained for other plants such as Arabidopsis with only nine (9) NLP proteins [63], 31 in *Brassica napus* [6], eight (8) in maize [64], six (6) in rice [65], and only five (5) in sorghum [31]. The high proportion of NLPs in cotton could be a pointer to their significant role in enhancing survival under nitrogen-deficient conditions. The tetraploid cotton genome, underwent whole-genome duplication (WGD), thus explaining the almost double the number of NLP proteins in *G. hirsutum*, a similar observation was also observed in the variation of the number of NLP proteins in the *Brassica* families in which *B. napus* harbored the highest number of NLP proteins. Moreover, the NLP proteins were found to be members of a dual domain, the RWP-RK, and PBI domains, though the PBI members were the dominant group.

The three cotton species NLP proteins were grouped into three as per the phylogenetic tree analysis, the classification of the cotton NLP proteins was coherent to previous studies in which three clades have been the dominant grouping number [6]. We further investigate to determine the evolution pattern of the cotton NLP proteins by determining the non-synonymous (dn) and synonymous (ds) substitution rate, majority of the paralogous gene pairs showed that their ds/dn ratios were less than 1, an indication that they underwent negative or purifying type of selection pressure [66]. The genetic fingerprint of an organism is contained in its DNA, and the processes by which DNA is translated using Ribonucleic acid (RNA) into a protein is called transcription and translation [67]. The transcription and translational process are rather quick and mistake are bound to occur in the process, non-lethal mistakes occur through synonymous mutation while lethal effects occur through non-synonymous mutation, being it results in to complete synthesis of new proteins or stops the process altogether [68]. However, non-synonymous type of mutation results in positive change, favoring the expression of genes, which in turn improves the adaptability of an organism to its environment, the negative/purifying selection could be due to the integral role played by the NLP proteins in cotton. Similar observations were made in the analysis of the late embryogenesis abundant (LEA) proteins, in which the majority of the proteins in all the eight subfamilies underwent a negative/purifying type of mutation [34].

The cotton *NLP* genes were found to be distributed in all the chromosomes, this was evident in both tetraploid and diploid types. The presence of the genes in all the chromosomes could perhaps explain their functions within the plant as an integral component in enhancing the plant’s adaptability to the N-limited environment. In addition, four subcellular compartments were predicted to harbor the proteins encoded by the cotton *NLP* genes, the highest proportions of the proteins were predicted to be located within the nucleus. The results were in agreement with previous reports in which the entire NLP proteins in *Brassica napus* except one were located within the nucleus [6], moreover, the nucleus plays an important role in protein formation. Nitrogen metabolism occurs partly in the plastids, the plastids exist in arrange of forms, from the photosynthetic chloroplast to the pigment-storing chromoplast of flowers and fruit, and the starch-accumulating amyloplasts found in storage organs. However, despite their varied forms, the plastids have a unique ability to transform from one form to another. Despite the existence of plastids in various forms, all plastids in a particular organism have common DNA, inherited from the maternal parent, showing that the nuclear genome is vital regulatory role over the plastid morphology and function [69]. Furthermore, the plastids are estimated to contain at least 1000, and possibly as many as 5000 different proteins, the vast majority of these proteins are encoded within the nucleus and their corresponding DNA sequences are believed to have migrated from the plastid genome during the course of evolution [70]. Thus, the high number of the cotton NLP proteins predicted to be sublocalized within the nucleus could possibly be explained by either duplication or transfer events from the plastids to the nucleus. Moreover, nitrogen utilization has been well investigated in yeasts and filamentous fungi and found to be regulated by a process known as the nitrogen catabolite repression (NCR), which modulates the gene expression through a family of GATA binding zinc finger transcription factors [71]. In *Saccharomyces cerevisiae*, when exposed to N-limited conditions, the two forms of NCR, *gln3* and *gat1* move to the nucleus and activate genes containing upstream GATA sites [72], thus the high prediction of the NLP proteins encoded by the *NLP* genes within the nucleus is an indication that these groups of proteins are integral in nitrogen assimilation and metabolism.

Even though several studies have linked the plant miRNAs to environmental stresses such as drought, cold, salt, and heat, new pieces of evidence have shown that miRNAs are important in enhancing plant’s adaptability to low nitrogen and phosphorus conditions [73]. Differential expression of a number of plant miRNAs has been observed in both dicotyledonous plants such as *Glycine max* [74] and *Arabidopsis thaliana* [75], similar, observations have been made among the monocotyledonous plants, maize [76] and rice [77]. In this investigation, a number of miRNAs were found to target various *NLP* genes in *G. hirsutum* and *G. raimondii*. Among the *NLP* genes obtained from the tetraploid upland cotton, *G. hirsutum*, ghr-miR167a and ghr-miR167b, were found to target four (4) genes each, which were, *Gh_A08G0810* (Protein unc-45 homolog B), *Gh_A13G2318* (tetratricopeptide repeat protein 1), *Gh_D08G0987* (protein unc-45 homolog B), and *Gh_D13G2470* (tetratricopeptide repeat protein 1). The same miRNA has been found to be targeted by two genes of the Auxin Response Factors (ARF transcription factors), *ARF6* and *ARF8* [78]. The ARF transcription factor, *ARF8* do regulate the development of the lateral roots, and its expression was induced in the pericycle and lateral root cap cells under N-limited conditions [79]. Further pieces of evidence have shown that overexpression of miR160 and miR167, do enhance lateral roots development under N-limited conditions, which is believed to improve or boost the capability of the plants to maximize the uptake of the little available nitrogen. The NAC plant’s transcription factors (TFs) are among the top-ranked plant stress-responsive TFs, the NACs consist of three gene families, *NAM* (No Apical Meristem), *ATAF* (Arabidopsis Transcription Activation Factor), *CUC* (CUp shaped Cotyledon) [80]. Five members of the *NAC* genes have been found to be targeted by miR164, including *NAC1* and *CUC1* genes, *NAC1* is integral in auxin signal transduction for the growth and development of lateral roots [81], while *CUC1* is important for the normal embryonic, vegetative, and floral development in plants [82]. The same miRNAs have been found to be upregulated in maize under N-limited conditions. In our investigation, miR164 was found to target four genes such as *Gh_A03G0443* (protein RKD4), *Gh_A12G0439* (serine/threonine-protein kinase *EDR1*), *Gh_D03G1095* (protein *RKD4*), and *Gh_D12G0440* (serine/threonine-protein kinase *EDR1*). The results showed that these genes could possibly be involved in enhancing lateral root development, and improving adaptability of cotton plant to N-limited environments. The miRNAs are an extensive class of ~22-nucleotide noncoding RNAs thought to regulate gene expression in metazoans, moreover, plants are often exposed to various stress factors and the defense mechanism devised by plants to cope with adverse climatic conditions is the reprogramming of gene expression by microRNAs (miRNAs) [83]. The high number of miRNA target on the various genes obtained for *G. raimondii* possibly could explain in part why various vital QTLs are often mapped on the D subgenome chromosomes as opposed to A subgenome chromosomes. Polyploidization contributed significantly to the alteration of the gene functions in the tetraploid cotton, moreover, higher number of transcription factors in tetraploid cotton are contributed by Dt subgenomes compared to the At subgenome, furthermore, previous reports have shown that a number of significant QTLs were mapped in Dt subgenome than At subgenome [84]. For instance, QTL mapping of drought and salt tolerance in an introgressed recombinant inbred line population of Upland cotton, 11 QTL were detected on 8 chromosomes, in which 10 of the QTLs were located on the D subgenome [85]. Moreover, in comprehensive Meta QTL analysis for fiber quality, yield, yield-related and morphological traits, drought tolerance, and disease resistance in tetraploid cotton, A subgenome harbored 536 QTL, while the D subgenome had 687 QTL [86]. Furthermore, a detailed analysis of the RFLP map revealed that a number of resistance genes (R genes) are located on the D subgenome chromosomes compared to A subgenome chromosomes, with five out of six R genes located on the D subgenome chromosomes [87]. Moreover, analysis of the genome structure of tetraploid cotton, *G. hirsutum* showed that 75% (125/166) of the polymorphic loci were tagged on the D-subgenome [88]. The merger of divergent genomes in a common nucleus has been argued to present a shift from genetic flexibility to genetic fixation providing a mechanism for response to selection [89]. These findings provide a unique contribution of the D subgenome to whole-genome evolution and adaptability of tetraploid cotton to its environment, and the high miRNA target could be playing a role in fine-tuning the expression of the D subgenome genes. A number of reasons have been forwarded, explaining the significance of the D subgenome in tetraploid cotton, a higher underlying mutation rate, a higher level of DNA polymorphism, and a non-homologous chromosome rearrangement [90].

Gene expression profiling is a powerful mechanism for studying biological processes, especially tissue/organ-specific ones, at the molecular level, and thus we carried out a detailed analysis of the upland cotton *NLP* genes in root, stem, and leaf tissues at three true leaf stage, under nitrogen-deficient condition. The results showed that the proportions of the *NLP* genes increased with increased exposure to N-limited conditions, more significantly, the novel gene *Gh_A05G3286* (NLP5) showed upregulation with increased exposure to N-limited conditions. The stress-responsive genes have been found to show higher expression with an increase in stress level and duration of exposure [91]. Moreover, gene induction has been found to be governed by time and the organ profiled [92]. The ability of the VIGS-plants to tolerant low nitrogen condition was highly compromised, being there was a significant reduction in root biomass, shoot biomass and leaf chlorophyll content. Furthermore, the antioxidant enzymes, proline, and SOD levels were significantly scaled-down, whereas the MDA level was significantly increased. The increased level of MDA and a significant reduction in SOD and proline indicated that the VIGS-plants were affected under N-limited conditions. The results were in agreement to previous findings in which the accumulation of abscisic acid (ABA) and H_2_O_2_ in leaves of rice, was found to be induced by N deficiency [93], moreover, overexpression of Arabidopsis *NLP7* gene improved plant growth while the nlp7 mutant plants growth showed constitutive N-deficient phenotypes on both nitrate-rich and limiting media [94]. The significant reduction in the growth of the VIGS-plants both under N-limited and none-limited conditions showed that the cotton *NLP* genes play an integral role in enhancing plant growth and development. When plants are exposed to any form of stress, either biotic or abiotic stress factors, the internal and external cell environments become altered, this is due to increased production of reactive oxygen species (ROS), shut down or slowdown of other cellular and biological processes and or release of reactive nitrogen species (RNS) which modify enzyme activity and gene regulation [95]. The VIGS-plants exhibited a higher level of malondialdehyde (MDA) and a significantly low level of leaf chlorophyll content. Moreover, it has been found that N-deficiency has a negative effect on the leaf chlorophyll content concentration [96]. Moreover, all three N-limited stress-responsive genes were all downregulated. PP2A is a regulatory subunit of the Tap46 an integral component of the target of rapamycin (TOR) signal pathway, the downregulation of the gene indicated that the TOR signal pathway was deactivated leading to repression of nitrogen mobilization. Moreover, the silencing of *Tap46* in tobacco-caused chromatin bridge formation at anaphase, which elucidated the integral role of the *Tap46* in plant growth and development as a component of the TOR signaling pathway [97].

## 4. Materials and Methods

### 4.1. Plant Materials and Treatments

Seeds of the upland cotton species, *G. hirsutum* (AD)_1_, accession number ZM-30014, was obtained from the cotton research institute seed bank. The seeds were germinated in the plant growth chambers with conditions programmed at 16h photoperiod 25/18 °C day/night and with 60% humidity. After three days of germination, the seedlings were transplanted into a hydroponic setup, in the greenhouse of the cotton research institute, Chinese Academy of Agricultural Sciences, CRI-CAAS, China. To analyze the expression patterns of the cotton *NLP* gene family members in different organs, the root (Rt), leaf (leaf), and stem (St) were profiled. To investigate the expression patterns of the cotton *NLP* genes under N-deficiency stress. The cotton seedlings were grown in sand and watered with nutrient solution [98]. Moreover, at three-leaf stages, the seedlings were subjected to N-limited conditions by watering with nutrient solution. The experimental unit was a set of three adjacent plants. Previous studies indicated that 0.05 mM NO_3_^−^ concentration in the nutrient solution provided an N-deficient supply to plants and 7.5 mM NO_3_^−^ provided an N-abundance supply [99]. The treatment nutrient solutions was a modified nutrient solution containing 0.05 mM or 7.5 mM of NO_3_^−^, with changes in NO_3_^−^ concentration balanced by changes in SO_4_^2^^−^; concentration [99]. The pH value of the nutrient solution was adjusted with KOH and H_2_SO_4_ and maintained between 5.8 to a maximum value of 6.4. The solution was regularly replaced after every three days. The plants were grown for 15 days. The samples were collected in three replicates after 0 h, 3 d, 6 d, 9 d, 12 d, and 15 d of treatment and stored under −80 °C for RNA extraction.

### 4.2. Identification and Protein Physiochemical Properties Analysis of the NLP Family Genes in Plants

Genomic, coding sequences (CDS) and protein sequences from three cotton species, *G. hirsutum*, *G. arboreum*, and *G. raimondii* together with other plants, *Arabidopsis thaliana*, *Theobroma cacao,* and *Brassica Rapa* were downloaded. The NLP proteins for *G. hirsutum* were obtained from the cotton research institute genome database (http://mascotton.njau.edu.cn). The *G. arboreum* NLP proteins were downloaded from the Beijing genome institute database (https://www.bgi.com/), and for the D genome *G. raimondii*, *T. cacao*, and *Glycine max* were retrieved from Phytozome 12.0 database (http://www.phytozome.net/), with *E*-value < 0.01, while for the model plant, *A. thaliana*, the NLP proteins were downloaded from the TAIR (http://www.arabidopsis.org). Since extensive research on Arabidopsis NLP proteins has been done, the protein sequences of Arabidopsis NLP proteins were used to carry out blast search for all the cotton and other plants NLP proteins. The redundant sequences were removed from further analysis. Moreover, PROSITE (http://prosite.expasy.org/scanprosite/) and SMART (http://smart.embl-heidelberg.de/) were used to confirm the presence of the NLP protein domain. SMART and PFAM database were used to verify the presence of the NLP-related domains, RWP-RK (PF02042) and Phox and Bem1 (PBl), Pfam number of PF00564, with an *E*-value cutoff of 1 × 10^−5^. Other protein’s physiochemical properties of the three cotton NLP proteins such as the isoelectric charge (*pI*), grand hydropathy values (GRAVY), the charge, molecular weights (MW) and other protein properties were estimated by ExPASy Server tool (http://www.web.xpasy.org/compute_pi/).

### 4.3. Phylogenetic Analysis of the NLP Family in Cotton Species with Other Plants

The obtained NLP protein sequences from the three cotton species, together with NLP proteins from *A. thaliana, T. cacao*, and *Vitis vinifera* were used to carry out a phylogenetic analysis of the NLP proteins in order to understand their evolution pattern. The proteins were aligned, by adopting multiple sequence alignment using ClustalW a component of MEGA 6 tool, the phylogenetic tree was then generated, by adopting neighboring joint (NJ) and Maximum Likelihood (ML) method with P distance Model, with 1000 bootstrap replications, site cutoff coverage of 100% and missing gaps/missing data treatment with complete deletion [32]. The generated tree was visualized by the use of Fig Tree version 1.4.0 (http://tree.bio.ed.ac.uk/software/figtree/). We further analyzed the coding sequences to calculate synonymous (ds) and non-synonymous (dn) rate of mutation by use of dn/ds _calculator version 2.0 and further validated by use of an online tool, Synonymous Non-synonymous Analysis Program (https://www.hiv.lanl.gov/content/sequence/SNAP/SNAP.html).

### 4.4. Chromosomal Locations, Subcellular Localization Prediction, and Gene Structure Analysis

The chromosomal locations of the three cotton species *NLP* genes were determined based on the results of the reciprocal BLASTP analysis, the cotton NLP homolog genes were determined with a threshold of > 80% in similarity and at least in 80% alignment ratio to their protein total lengths. MapChart tool (https://mapchart.net/) was employed to plot the *NLP* genes in the various cotton chromosomes by use of their physical position in base pairs (bp). The subcellular predictions of the proteins encoded by the three cotton *NLP* genes were predicted by the use of an online tool Wolfsport (https://www.Wolfpsort.hgc.jp/). The results were further validated by the use of TargetP1.1 (http://www.cbs.dtu.dk/services/TargetP/) server and Protein Prowler Subcellular Localisation Predictor version 1.2 (http://www.bioinf.scmb.uq.edu.au/pprowler_webapp_1-2/). Finally, the gene structures were analyzed by the use of a gene structure displayer server (http://gsds.cbi.pku.edu.cn).

### 4.5. Cis-regulatory Element Analysis and the miRNA Target Prediction of the Cotton NLP Proteins

The promoter sequences of 1 kb up and downstream of the coding sequence regions were analyzed to obtain the possible *cis-*regulatory elements by employing an online tool the Plant CARE (http://bioinformatics.psb.ugent.be/webtools/plantcare/html/). The results were further validated by the use of the PLACE database (http://www.dna.affrc.go.jp/PLACE/signalscan.html). Moreover, we employed the use of the miRNA database in order to determine the possible miRNA targeting the various cotton NLP proteins, the CDS for various cotton species were uploaded onto the online tool the psRNATarget server with default parameters (http://plantgrn.noble.org/psRNATarget/).

### 4.6. RNA Isolation and Quantitative Reverse-Transcription PCR

Total RNA was extracted from the plant organs using an RNA Extraction Kit obtained from Aid Lab, The RNA sample quality was evaluated by use of the gel electrophoresis and a NanoDrop 2000 spectrophotometer, only RNAs with specification criterion of 260/280 ratio of 1.8–2.1, 260/230 ratio ≥ 2.0 were retained and used for further analysis. The cDNA was synthesized from 1 µg of total RNA using M-MLV transcriptase (TaKaRa Biotechnology, Dalian, China) as described within the manufacturer’s user instructions manual. To determine the organ-specific expression profiles of *GhNLP* genes, publicly available *G. hirsutum* RNA-seq data available at cotton functional genome database (https://cottonfgd.org/analyze/) were used to quantify the expression levels of the *G. hirsutum* genes in different organs as fragments per kilobase (Kb) of exon per million reads mapped (FPKM) values using Cufflinks with default parameters. Two independent biological replicates were analyzed per sample. Fifty-three upland cotton *NLP* genes were selected based on the phylogenetic tree and gene structure analysis, for the RT-qPCR validation, the same method had been applied by Magwanga et al. [34], in the selection of the *LEA* genes for RT-qPCR analysis. Reverse transcription-quantitative polymerase chain reaction (RT-qPCR) was performed on a Bio-Rad CFX96 Real-Time System (USA) according to a previously described method [10]. The BatchPrimer3 Primer software (http://batchprimer3.bioinformatics.ucdavis.edu/) used to synthesize the *GhNLP* specific primers (Appendix A). Each reaction was carried out in three biological replicates in a reaction volume of 20 µl containing 1.6 µL of gene-specific primers (1.0 µM), 2.0 µL of cDNA, 10 µL of SYBR green, 0.2 µl of ROX Reference Dye II, and 6 µl of sterile distilled water. The PCR program was as follows: 95 °C for 3 min, 45 cycles of 10 s at 95 °C and 30 s at 60 °C, and then melt curve 65 °C to 95 °C, increment 0.5 °C for 5 s. Melting curves were generated to estimate the specificity of these reactions. Relative expression levels were calculated using the 2^−∆∆*C*t^ method, with *GhActin* used as internal controls [100].

### 4.7. Generation of Transiently Transformed G. hirsutum Plants with Repression of Gh_A05G3286 (NLP5)

The *NLP* gene, *Gh_A05G3286 (NLP5)* with a sequence length of 819 bp fragment was amplified from cDNA using the VIGS-F (5′ACACGTGCTTGGACTCTGTC3′) and VIGS-R (5′CGAATTTGAT GTCAGCGCGT3′) primers. The fragment was amplified using KAPA HiFi HotStart ReadyMix, cloned into the pTRV2 vector to yield pTRV2:*Gh_A05G3286 (NLP5)* and verified through sequencing. The construction of the pTRV2 silencing vector and method of viral inoculation followed the protocol by Scofield et al. [43]. Viral controls included pTRV2:00, which is derived from the empty pTRV2 vector, and pTRV2: PDS that included a transcript that targets the phytoene desaturase (*PDS*) gene and acts as a visual marker of correct viral reconstitution. *G. hirsutum* seedlings were treated with the virus constructs to determine the effect of silencing *Gh_A05G3286 (NLP5).* VIGS-infiltrated seedlings were left to grow for 14 days, upon the emergence of two true leaves, the leaves were harvested and stored at −80 °C for RNA extraction to confirm the VIGS efficiency. Seedlings were then simultaneously subjected to N-deficiency, stress treatment by growing in the sand (nutrient-deficient rooting material), and were watered with two different nutrient solutions with NO_3_^−^ concentrations of 0.05 mM and 7.5 mM to provide N-deficient and N- abundant supply, respectively. The experimental unit was a set of three adjacent plants. Previous studies indicated that 0.05 mM NO_3_^−^ concentration in the nutrient solution provided an N-deficient supply to plants [101] and 7.5 mM NO_3_^−^ provided an N-abundant supply [99]. The treatment nutrient solutions were modified nutrient solution containing 0.05 or 7.5 mM NO_3_^−^, with changes in NO_3_^−^ concentration balanced by changes in SO_4_^2−^ concentration [99]. The pH value of the nutrient solution was adjusted with KOH and H_2_SO_4_ and maintained between 5.8 to a maximum value of 6.4. The plants were irrigated with a nutrient solution every three days. The plants were grown for 15 days.

### 4.8. Evaluation of Physiological, Morphological, Biochemical Traits and Expression Analysis of the N-Limited Responsive Genes

We evaluated various physiological and morphological parameters such as chlorophyll content, shoot biomass and root biomass was measured. The chlorophyll content was evaluated through the non-destructive method [102], while shoot and root biomass were measured by the use of a weighing balance in grams. Three biological replicates and three technical replicates were performed for all the measurements. Moreover, we evaluated the concentration of the antioxidant and oxidant enzymes in the leaves of the VIGS and the non-VIGS cotton plants grown under N-limited conditions. The superoxide dismutase (SOD) activity was evaluated by monitoring the inhibition of the photochemical reduction of nitro blue tetrazolium (NBT), as described by Giannopolitis and Ries though with slight modifications [103] and finally proline was evaluated by the method described by Van Assche [104]. Malondialdehyde (MDA) was evaluated between the two plants, wild types and the VIGS-plants under the three stress levels. MDA was determined as a measure of lipid peroxidation [105]. Moreover, three N-limited stress-responsive genes were profiled in order to evaluate their expression levels on the VIGS-plants and the wild types under N-limited conditions. The cotton 60S ribosomal protein L18A (*GhRPL18A*), with forward sequence “TTTCCGGTTTCACCAGTACC”, and the reverse sequence “AACAAGTTGGGTTTCGATGC”, cotton PP2A regulatory subunit (*GhTap46*), with forward sequence “GGCTGCAGAGGCAAAA CTTA), and reverse sequence “CTTCTCTTGGGCAGCATCA T” [50], cotton cytochrome P450 CYP78A5 monooxygenase (*KLU*), with forward sequence “GCTGATAGGCCAGTGAAGGA” and the reverse sequence “TCTTGAGCCTTTGCTTGGAT” [51], were profiled on the leaf tissues of *Gh_A05G3286 (NLP5)-*silenced plants, wild type (WT) and the positive controlled plants (TRV2:00), grown under N-limited conditions. All evaluations were carried after 15 days of growth under N-limited conditions, grown in sand and watered with a modified nutrient solution. The cotton *GhActin* was used as the internal controls.

## 5. Conclusions

The loss due to excessive use of nitrogenous fertilizer is estimated at 2–20% through evaporation, 15–25% get fixed by reacting with the organic compounds in the clay soil and the remaining 2–10% is lost through runoff and groundwater which eventually end up into the water bodies [106]. Based on the effects and the price of the nitrogenous fertilizers, a different approach is needed by plant breeders not only to develop nitrogen-use-efficient cotton germplasm but also to develop more genotypes that are resilient with higher adaptability to N-limited conditions. In this research work, we found 226 *NLP* genes in the three cotton species, with 105, 61, and 60 genes in *G. hirsutum*, *G. raimondii,* and *G. arboreum*, respectively. The proteins encoded by the *NLP* genes were classified into three clades and were found to possess a wider genome distribution, covering all the cotton chromosomes. Deep Insilico analysis of the proteins encoded by the cotton *NLP* genes showed that over 50% of the proteins were predicted to be sublocalized in the nucleus. The nucleus regulates all the cellular activities, and the presence of the NLP proteins in the nucleus could perhaps explain their critical role in enhancing plants ability to tolerate low or nitrogen deficient conditions. Moreover, the GRAVY values of most the proteins encoded by the *NLP* genes was less than one, an indicated that these proteins are hydrophilic in nature. Hydrophilicity is a property associated with a number of proteins encoded by various stress-responsive genes such as the *LEA* genes [34]. Moreover, the functional characterization of the novel gene through VIGS approached, revealed that the VIGS-plant’s ability to tolerate N-limited conditions was significantly reduced compared to their wild types. Furthermore, the VIGS-plants registered a significant reduction in chlorophyll content, fresh shoot biomass, and fresh root biomass compared to the WT under N-limited condition. Moreover, evaluation of oxidant and antioxidant enzymes showed that the VIGS-plants had a higher concentration level of oxidant enzyme, MDA but significant reduction in antioxidant enzymes, proline, and SOD compared to the wild types under N-limited conditions. Finally, profiling of the N-limited responsive genes, *GhTap46, GhRPL18A*, and *GhKLU* exhibited significant reduction on the tissues of the VIGS-plants, but were upregulated on the tissues of the wild types under N-limited condition. These results showed that the cotton NLP proteins encoded by the *NLP* genes could be playing a significant role in enhancing the tolerance level of cotton to N-limited conditions.

## Figures and Tables

**Figure 1 ijms-20-04953-f001:**
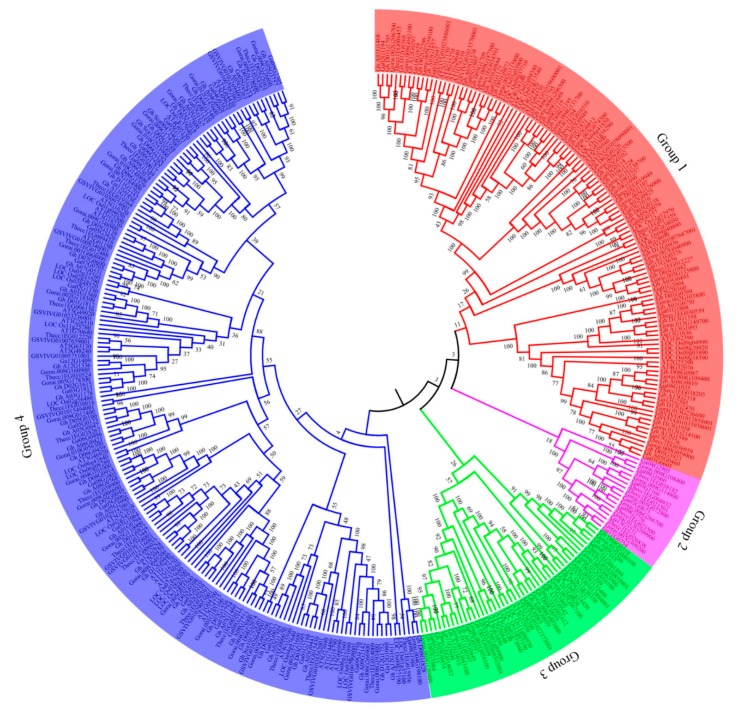
Phylogenetic tree analysis of the proteins encoded by the *NLP* genes in cotton and other plants. Different colours represent different groups, red: Group 1, purple: Group 2, green: Group 3′ and blue: Members of group 4. AT: *Arabidopsis thaliana*, LOC: *Oryza sativa*, Thecc: *Theobroma cacao*, GSVIVG: *Vitis vinifera*, Gorai: *Gossypium raimondii*, Ga: *Gossypium arboreum* and Gh: *Gossypium hirsutum.*

**Figure 2 ijms-20-04953-f002:**
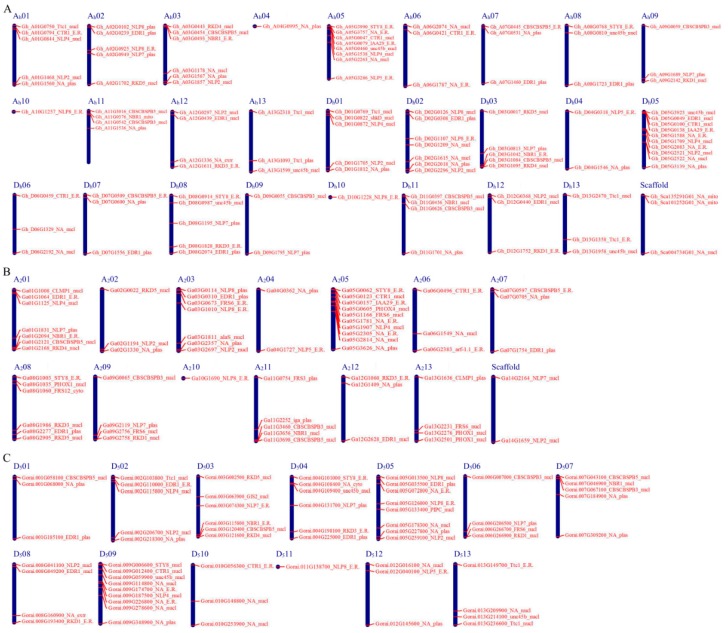
Chromosome mapping of the cotton *NLP* genes. (**A**) *NLP* genes mapped on *G. hirsutum* chromosomes. (**B**) *NLP* genes for diploid cotton, *G. arboreum* of the A genome. (**C**) *NLP* genes for *G. raimondii* diploid cotton of the D genome.

**Figure 3 ijms-20-04953-f003:**
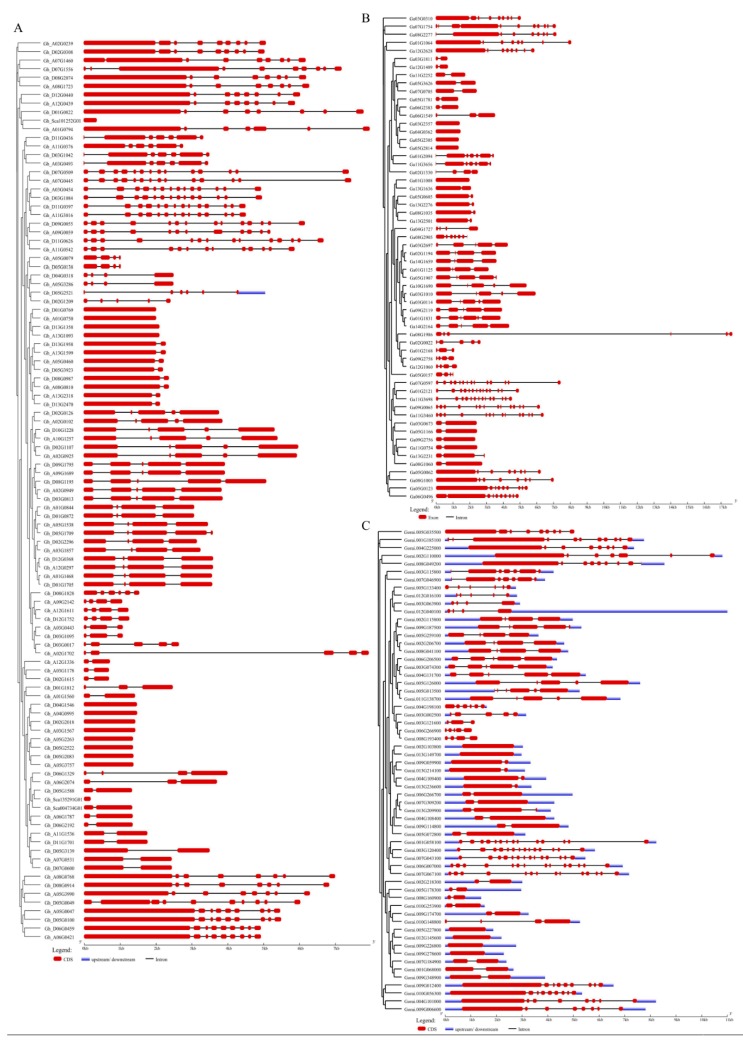
Cotton *NLP* gene structures. (**A**) Gene structures for the upland cotton *G. hirsutum*. (**B**) Gene structures for the *G. arboreum.* (**C**) Gene structural analysis of the *NLP* genes of *G. raimondii.*

**Figure 4 ijms-20-04953-f004:**
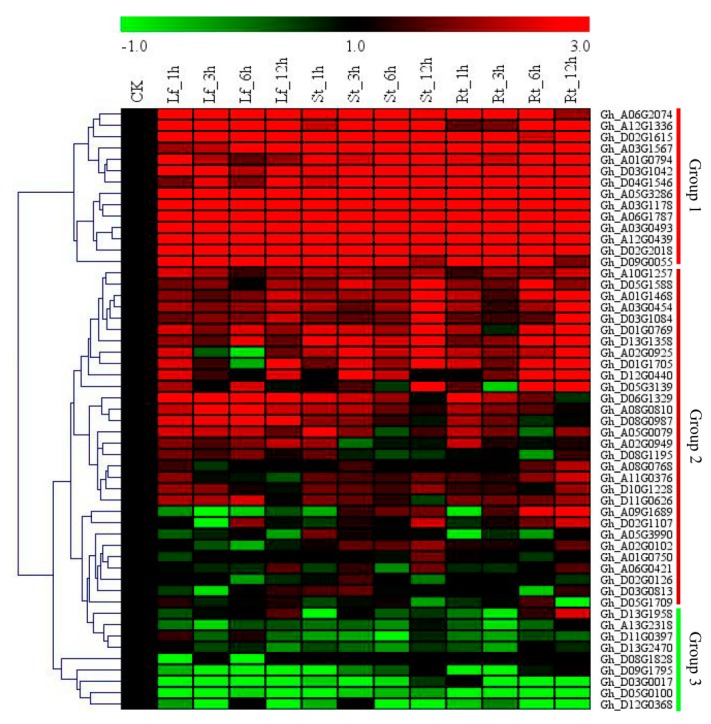
Expression Analysis of the selected *GhNLP* genes. Cotton seedlings were grown in a hydroponic setup with a modified Hoagland nutrient solution, N-limited condition imposed by adjusting the N-content to 0.05 mM NO_3_^−^ concentration, and normal N-condition maintained at 7.5 mM NO_3_^−^ concentration. The leaf, stem and root tissues were harvested at 0 h, 1 h, 3 h, 6 h, and 12 h. RNA extracted and the expression levels of the 53 selected *GhNLP* genes analysed by RT-qPCR with *GhActin* as the internal control. The heat map was visualized using Mev.exe program (Showed by log 2 values). Red: Up-regulated, green: Down-regulated and black-no significant difference in expression levels compared to control (ANOVA, *p* < 0.05).

**Figure 5 ijms-20-04953-f005:**
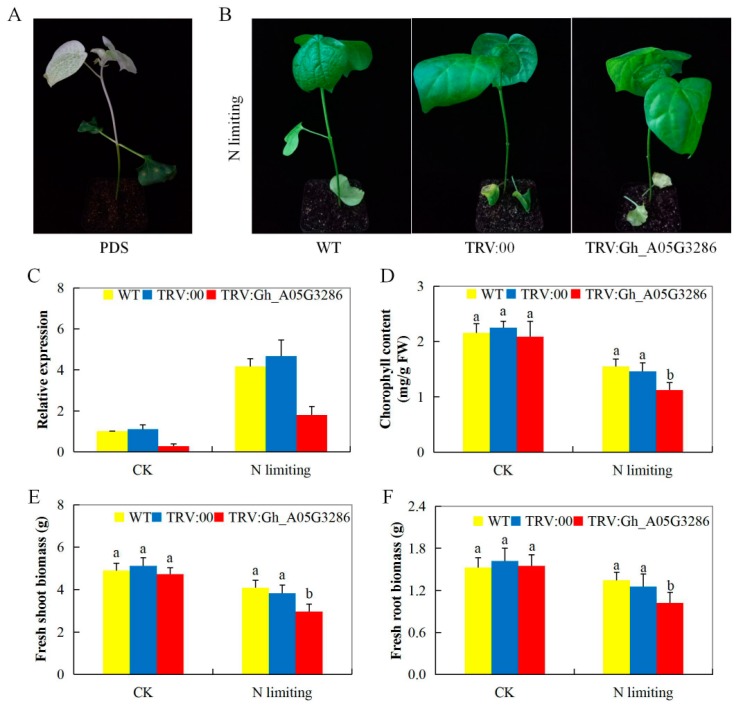
Phenotype observed in the silenced plants with the TRV: 00 empty vector, wild type plants and *Gh_A05G3286 (NLP5)*-silenced plants at 12 days post-inoculation. (**A**) Albino appearance on the leaves of the PDS infused plants. (**B**) Photograph of the plants taken after 15 days of N-deficiency exposure. (**C**) RT-qPCR analysis of the change in the expression level of the *Gh_A05G3286 (NLP5)* gene in cotton plants treated with VIGS. (**D**–**F**) Evaluation of chlorophyll, fresh shoot biomass and fresh root biomass, “TRV:00” represents the plants carrying control the TRV2 empty vector, “TRV: *Gh_A05G3286 (NLP5)*” represents the *Gh_A05G3286 (NLP5)-*silenced plants. Letters a/b indicate statistically significant differences (two-tailed, *p <* 0.05). The error bars of the *Gh_A05G3286 (NLP5)* gene expression level represent the standard deviation of three biological replicates.

**Figure 6 ijms-20-04953-f006:**
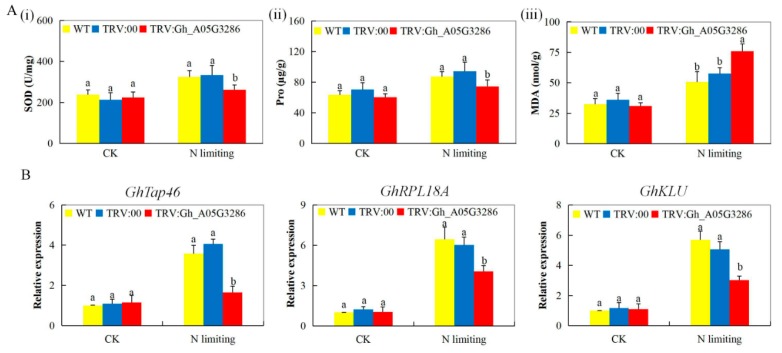
Evaluation of biochemical components, oxidant and antioxidant enzyme concentration levels in *Gh_A05G3286 (NLP5)*. (**A**) (i): Quantitative determination of SOD concentration. (**ii**): Quantitative determination of proline concentration. (iii): Quantitative determination of MDA concentration (**B**) (i–iii): Stress responsive transcription analysis of the various plants exposed to N-limited conditions for 15 days. Letters a/b indicate statistically significant differences (two-tailed, *p <* 0.05). The *GhActin* gene was used as internal control. The error bars of the *Gh_A05G3286 (NLP5)* gene expression level represent the standard deviation of three biological replicates.

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
