# Peer review of "Map-Based Functional Analysis of the GhNLP Genes Reveals Their Roles in Enhancing Tolerance to N-Deficiency in Cotton"

_ijms, 2019, doi:10.3390/ijms20194953_

Round 1
Reviewer 1 Report
Review of IJMS-543632
A Novel GhNLP Genes as Identified within the QTL Region Reveal Their Role in Enhancing N-deficiency Tolerance in Cotton
Richard Odongo Magwanga, Joy Nyangasi Kirungu, Pu Lu, Xiaoyan Cai, Zhongli Zhou, Yanchao Xu, Yuqing Hou, Kunbo Wang and Fang Liu
The authors used QTL mapping by sequencing to identify 226 nodule-inception-like protein (NLP) genes in cotton (Gossypium spp.). Of these, 105 were identified in Gossypium hirsutum, 61 were identified in G. raimondii and 60 were identified in G. arboreum, respectively. They then performed bioinformatics analyses to construct a phylogeny of the NLP proteins, map their chromosomal positions, infer their molecular weights, isoelectric points and hydropathy values, and to identify miRNAs and cis-regulatory elements associated with these genes. Using RT-qPCR analysis they identified five of these genes that were significantly up-regulated under N-limited conditions. They used virus-induced gene silencing (VIGS) to knock down the expression of Gh NLP5, and showed that this compromised the ability of these plants tolerate N-limited conditions. Expression of the N-stress responsive genes, GhTap46, GhRPL18A, and GhKLU was also significantly down-regulated in these plants under N-limited conditions. These results suggest that Gh NLP5 plays an important role in tolerating N-deficiency in cotton
Overall, the analysis of the gene family provides useful information about the proteins and their evolution. The qRT-PCR and VIGS analyses also provide clues as to their potential roles in responses to N-deprivation and identify a candidate gene to use for breeding improved tolerance of N-limited conditions.
I therefore feel that this paper provides useful information worthy of publication after it has been thoroughly edited.
One major criticism is that the authors take it for granted that the readers will know the evolution of G. hirsutum. I suspect that this is not the case for many readers. I therefore suggest that they explain in the introduction that G. hirsutum is thought to be a tetraploid containing both an A and a D genome. This then explains why they chose to study G. arboretum, which has the A genome, and G. raimondii which has the D genome.
In this context, I would also like to see them discuss further how G. hirsutum seems to have lost 16 NLP genes relative to the diploid varieties (If one genome has 61 copies and the other has 60, why does G. hirsutum only have 105?) Can they identify which genes were lost? Better still, can they identify how they were lost?
This also brings up another very interesting finding which they mention then neglect (line 249): Why do significantly higher numbers of miRNA target the D genome in G. hirsutum ? Does this mean this is the main genome used in G. hirsutum? Or is this a way to shut down its expression? I would like to see the authors discuss this finding in much greater depth in their discussion. What does this tell us about the evolution of tetraploids?
The abstract is too long, yet should also indicate that they also performed a phylogenetic analysis of the NLP proteins, mapped their chromosomal positions, intron-exon structures and conducted ds/dn analysis showing that most underwent negative or purifying selection.
The English is mostly understandable, but nearly every sentence, starting with the title, contains errors that require attention. I recommend editing by a native English speaker.
More specific comments
1) The title contains numerous errors: A Novel GhNLP Genes as Identified within the QTL Region Reveal Their Role in Enhancing N-deficiency Tolerance in Cotton
· “A novel” is singular, whereas “GhNLP Genes as Identified within the QTL Region Reveal Their Role in Enhancing N-deficiency Tolerance in Cotton” is plural
· what is “the QTL region?” Please explain better!
2) Numerous acronyms are introduced without explanation. For example, on line 27, NLP should be explained the first time that it is introduced. Similarly, on line 45, MDA should be explained the first time it is introduced, especially since they explain superoxide dismutase (SOD) in the next line. Line 278 should explain PDS the first time it is mentioned.
3) Line 37 and throughout the text: “N-limiting condition” should be “N-limited conditions”
4) The materials and methods used in many experiments, and the rationale for using them, are described in the methods section rather than when they are first introduced. For example, Line 138 should provide a quick overview of the methods used to identify NLP proteins, line 152 should provide a quick overview of the methods used to classify the NPL proteins, line 176 should quickly state how they were mapped, line 203 should state how introns were identified, line 227 should summarize how cis-regulatory sequences were identified, line 240 should summarize how miRNA binding sites were identified, line 267 should summarize how gene expression was measured and lines 278 to 280 should summarize the rationale for using PDS and the structures and purposes of the VIGS constructs.
5) The authors should be careful to indicate that the protein localization and miRNA binding sites are based on computer simulations rather than experimental validation.
6) I would like the authors to provide more insight into the cotton NLP genes that were homologous to NLP proteins in other plants. Perhaps a table listing the cotton NLP genes that had orthologs in other plant species, and the known functions of these orthologs. This could also identify candidate genes for breeding programs to improve tolerance of N-limited conditions in cotton.
Line 59: many higher plants besides the leguminous plants can form associations with N-fixing bacteria, e.g. Gunnera! Please rephrase to explain that the leguminous plants are the most common or the most important group of plants that can form associations with N-fixing bacteria
Line 156: if the NLP protein classification was incongruent to previous studies this would imply that it differed and the NLP proteins are not highly conserved.
Line 158: is T. cacao really the closest relative, or just the closest relative in the other species that were analyzed? Please clarify!
Line 273 : Caption to figure 4 should indicate gene used as reference for qRT-PCR and numbers of biological and technical replicates.
Line 344 : Caption to figure 6 should indicate gene used as reference for qRT-PCR .
Author Response
English language and style
(x) Extensive editing of English language and style required
Response: We have carried out intensive language editing and corrected all the grammatical and syntax errors.
Yes | Can be improved | Must be improved | Not applicable | |
Does the introduction provide sufficient background and include all relevant references? | ( ) | ( ) | (x) | ( ) |
Response: Introduction section has been improved by adding information in relation to the evolution of tetraploid cotton, G. hirsutum is a tetraploid cotton of AD genome, which emerged as a result of polyploidization or whole genome duplication between G. raimondii and G. arboreum both diploid cotton of A and D genomes, respectively. | ||||
Is the research design appropriate? | (x) | ( ) | ( ) | ( ) |
Response: Thanks | ||||
Are the methods adequately described? | ( ) | (x) | ( ) | ( ) |
Response: in areas which were vaguely described have been well expounded and made easier for reproducibility of the research work by other researchers | ||||
Are the results clearly presented? | ( ) | ( ) | (x) | ( ) |
Response: Modification done as advised, in which in every section, a brief statement on how the___14 results were obtained needed to be mentioned, all done. | ||||
Are the conclusions supported by the results? | ( ) | (x) | ( ) | ( ) |
Response: Logically stated and give a comprehensive summary of the research findings | ||||
Comments and Suggestions for Authors
The authors used QTL mapping by sequencing to identify 226 nodule-inception-like protein (NLP) genes in cotton (Gossypium spp.). Of these, 105 were identified in Gossypium hirsutum, 61 were identified in G. raimondii and 60 were identified in G. arboreum, respectively. They then performed bioinformatics analyses to construct a phylogeny of the NLP proteins, map their chromosomal positions, infer their molecular weights, isoelectric points and hydropathy values, and to identify miRNAs and cis-regulatory elements associated with these genes. Using RT-qPCR analysis, they identified five of the genes that were significantly up-regulated under N-limited conditions. They used virus-induced gene silencing (VIGS) to knock down the expression of Gh NLP5, and showed that this compromised the ability of these plants tolerate N-limited conditions. Expression of the N-stress responsive genes, GhTap46, GhRPL18A, and GhKLU was also significantly down-regulated in these plants under N-limited conditions. These results suggest that Gh NLP5 plays an important role in tolerating N-deficiency in cotton
Overall, the analysis of the gene family provides useful information about the proteins and their evolution. The qRT-PCR and VIGS analyses also provide clues as to their potential roles in responses to N-deprivation and identify a candidate gene to use for breeding improved tolerance of N-limited conditions.
Response: We are humbled by the overall assessment of the manuscript. We do appreciate so much.
I therefore feel that this paper provides useful information worthy of publication after it has been thoroughly edited.
Response: We have carried intensive editing and re-writing of various sections as shown on the manuscript in order to improve the quality of the manuscript.
One major criticism is that the authors take it for granted that the readers will know the evolution of G. hirsutum. I suspect that this is not the case for many readers. I therefore suggest that they explain in the introduction that G. hirsutum is thought to be a tetraploid containing both an A and a D genome. This then explains why they chose to study G. arboretum, which has the A genome, and G. raimondii which has the D genome.
Response: It is true that the information provided was so much limited to cotton scientist, being the evolution of tetraploid cotton is a known fact. But we have made adjustments on the introduction in order to cover wider audience. The evolution and emergence of AD genome has been added within the introduction section.
In this context, I would also like to see them discuss further how G. hirsutum seems to have lost 16 NLP genes relative to the diploid varieties (If one genome has 61 copies and the other has 60, why does G. hirsutum only have 105?) Can they identify which genes were lost? Better still, can they identify how they were lost?
Response: It is a complex issue why the overall number of the genes in AD do not match the sum total of the genes in the two diploid cotton, A and D genomes. I have elaborated extensively and referenced various findings to support my argument.
This also brings up another very interesting finding which they mention then neglect (line 249): Why do significantly higher numbers of miRNA target the D genome in G. hirsutum ? Does this mean this is the main genome used in G. hirsutum? Or is this a way to shut down its expression? I would like to see the authors discuss this finding in much greater depth in their discussion. What does this tell us about the evolution of tetraploids?
Response: miRNA-gene interactions. The MicroRNAs (miRNAs) are an extensive class of ∼22-nucleotide noncoding RNAs thought to regulate gene expression in metazoans (Reinhart et al., 2002). Research has shown that plants do have their own defense mechanism against various stresses, for instance, drought is a normal and recurring climate feature in most parts of the world and plays a major role in limiting crop productivity. However, plants have their own defense systems to cope with adverse climatic conditions. One of these defense mechanisms is the reprogramming of gene expression by microRNAs (miRNAs) (Ferdous, Hussain & Shi, 2015). This implies that the miRNAs are integral for gene function, and the high number of miRNAs targeting the genes of G. raimondii, of the D genome provides more evidence why most of the vital QTLs are often mapped on the D subgenome chromosomes as opposed to A subgenome chromosomes in the tetraploid cotton.
The abstract is too long, yet should also indicate that they also performed a phylogenetic analysis of the NLP proteins, mapped their chromosomal positions, intron-exon structures and conducted ds/dn analysis showing that most underwent negative or purifying selection.
Response: Added
The English is mostly understandable, but nearly every sentence, starting with the title, contains errors that require attention. I recommend editing by a native English speaker.
Response: Extensive revision done.
More specific comments
1) The title contains numerous errors: A Novel GhNLP Genes as Identified within the QTL Region Reveal Their Role in Enhancing N-deficiency Tolerance in Cotton
· “A novel” is singular, whereas “GhNLP Genes as Identified within the QTL Region Reveal Their Role in Enhancing N-deficiency Tolerance in Cotton” is plural
· what is “the QTL region?” Please explain better!
Response: True, the title has been modified by deleting the article “A” , thus reads “A Novel GhNLP Genes as Identified within the QTL Region Reveal Their Role in Enhancing N-deficiency Tolerance in Cotton”.
2) Numerous acronyms are introduced without explanation. For example, on line 27, NLP should be explained the first time that it is introduced. Similarly, on line 45, MDA should be explained the first time it is introduced, especially since they explain superoxide dismutase (SOD) in the next line. Line 278 should explain PDS the first time it is mentioned.
Response: Added as advised.
3) Line 37 and throughout the text: “N-limiting condition” should be “N-limited conditions”
Response: All changed
4) The materials and methods used in many experiments, and the rationale for using them, are described in the methods section rather than when they are first introduced. For example, Line 138 should provide a quick overview of the methods used to identify NLP proteins, line 152 should provide a quick overview of the methods used to classify the NPL proteins, line 176 should quickly state how they were mapped, line 203 should state how introns were identified, line 227 should summarize how cis-regulatory sequences were identified, line 240 should summarize how miRNA binding sites were identified, line 267 should summarize how gene expression was measured and lines 278 to 280 should summarize the rationale for using PDS and the structures and purposes of the VIGS constructs.
Response: Added
5) The authors should be careful to indicate that the protein localization and miRNA binding sites are based on computer simulations rather than experimental validation.
Response: Indicated.
6) I would like the authors to provide more insight into the cotton NLP genes that were homologous to NLP proteins in other plants. Perhaps a table listing the cotton NLP genes that had orthologs in other plant species, and the known functions of these orthologs. This could also identify candidate genes for breeding programs to improve tolerance of N-limited conditions in cotton.
Response: Technically all the cotton NLP genes formed no ortholog gene pairs to any of the plant used, I have tried to redo the alignment, or blast search but the identity percentage is very low, thus limiting the chances of getting any ortholog gene pairs.
Line 59: many higher plants besides the leguminous plants can form associations with N-fixing bacteria, e.g. Gunnera! Please rephrase to explain that the leguminous plants are the most common or the most important group of plants that can form associations with N-fixing bacteria
Response: changed
Line 156: if the NLP protein classification was incongruent to previous studies this would imply that it differed and the NLP proteins are not highly conserved.
Response: Sorry, for the wrong use of the term “incongruent” it should have been congruent, correction done
Line 158: is T. cacao really the closest relative, or just the closest relative in the other species that were analyzed? Please clarify!
Response: Technically T. cacao was the closest relative of genus Gossypium compared to all the plants used in the analysis, changes made.
Line 273 : Caption to figure 4 should indicate gene used as reference for qRT-PCR and numbers of biological and technical replicates.
Response: Added
Line 344 : Caption to figure 6 should indicate gene used as reference for qRT-PCR .
Response: Added
Ferdous J, Hussain SS, Shi BJ. 2015. Role of microRNAs in plant drought tolerance. Plant Biotechnology Journal 13:293–305. DOI: 10.1111/pbi.12318.
Han Z, Liu Y, Deng X, Liu D, Liu Y, Hu Y, Yan Y. 2019. Genome-wide identification and expression analysis of expansin gene family in common wheat (Triticum aestivum L.). BMC Genomics 20. DOI: 10.1186/s12864-019-5455-1.
Li G, Hou M, Liu Y, Pei Y, Ye M, Zhou Y, Huang C, Zhao Y, Ma H. 2019. Genome-wide identification, characterization and expression analysis of the non-specific lipid transfer proteins in potato. BMC Genomics 20. DOI: 10.1186/s12864-019-5698-x.
Reinhart BJ, Weinstein EG, Rhoades MW, Bartel B, Bartel DP. 2002. MicroRNAs in plants. Genes and Development 16:1616–1626. DOI: 10.1101/gad.1004402.
Reviewer 2 Report
The manuscript by Magwanga et al has described the NLP proteins. Howevre it lack clarity to evaluate. Here I presented my concerns.
Please find them and correct it
1. Report regarding molecular mass and isoelectric point in the abstract section looking ackward. Need to remove it.
2. There are several typo errors. Please correct them.
3. Line 161: what is the ds/dn ratio?
4. The figures are so poor to evaluate. Kindly provide resolved photographs. So that reviewer can evaluate them.
5. Figure 5: statistical analysis is missing.
6. How did author knew the phylogenetic tree should be constructed using Neighbor joining approach? They should have run analysis for model selection before construction of the phylogenetic tree.
7. It need clear photographs before making any conclusive decision on this manuscript.
Author Response
Open Review
(x) I would not like to sign my review report
( ) I would like to sign my review report
English language and style
( ) Extensive editing of English language and style required
(x) Moderate English changes required
Response: Thanks, corrections have been made and various syntax and grammatical errors corrected.
( ) English language and style are fine/minor spell check required
( ) I don't feel qualified to judge about the English language and style
Yes | Can be improved | Must be improved | Not applicable | |
Does the introduction provide sufficient background and include all relevant references? | ( ) | ( ) | (x) | ( ) |
Response: missing gaps have been adjustments in order to provide comprehensive information not only limited to cotton researchers but to a broader audience. | ||||
Is the research design appropriate? | ( ) | (x) | ( ) | ( ) |
Response: Thanks, adjustment have been dome to make the experiment design easier for reproducibility by future researchers | ||||
Are the methods adequately described? | ( ) | ( ) | (x) | ( ) |
Response: Missing gaps added and well explained. | ||||
Are the results clearly presented? | ( ) | ( ) | (x) | ( ) |
Response: Thanks, we have made various adjustments | ||||
Are the conclusions supported by the results? | ( ) | ( ) | (x) | ( ) |
Response: Logically stated and give a comprehensive summary of the research findings | ||||
Comments and Suggestions for Authors
The manuscript by Magwanga et al has described the NLP proteins. Howevre it lack clarity to evaluate. Here I presented my concerns.
Please find them and correct it
1. Report regarding molecular mass and isoelectric point in the abstract section looking ackward. Need to remove it.
Response: summarized and the values deleted.
2. There are several typo errors. Please correct them.
Response: thorough reviewed done and all the typo errors corrected
3. Line 161: what is the ds/dn ratio?
Response: In the analysis of the selected type, we evaluated the synonymous (ds) and non-synonymous (dn) rate of substitution. The ds/dn ratio is significant in evaluating the type of selection pressure, which acted on the proteins encoded by the NLP genes, the ds/dn value of less than 1 indicates beneficial selection, those that are less than 1 signify negative selection or purifying selection and those of unit value shows that the mutational effect was neutral [32].
4. The figures are so poor to evaluate. Kindly provide resolved photographs. So that reviewer can evaluate them.
Response: Thanks, I have provided both PDF and TIFF format for all the figures.
5. Figure 5: statistical analysis is missing.
Response: Technically RT-qPCR expression analysis, do not require the signs for significance expression, thus no labels or asterisks used on figure 5C, the figure elaborates on the variation on expression levels of the knocked gene. The RNAi method only suppresses the expression but do not completely mask the expression of the target gene.
6. How did author knew the phylogenetic tree should be constructed using Neighbor joining approach? They should have run analysis for model selection before construction of the phylogenetic tree.
Response: in order to understand the evolution and emergence of various proteins encoded by the plants functional genes, the use of phylogenetic tree provides the very foundation, thus enables the detection of ortholog or paralog gene pairs. We have applied the phylogenetic tree analysis in order to understand the evolution nature by comparing the cotton NLP genes with other plants. Moreover, several investigations have been done by applying the phylogenetic tree analysis. For instance, to better understand the evolutionary relationship among nsLTPs, a phylogenetic analysis of nsLTPs identified in potato, Arabidopsis, and rice was performed using ClustalX2.1 and MEGA7 software (Li et al., 2019).
Through blast search against the Triticum aestivum genome database from GRAMENE (http://ensembl.gramene.org/), 241 wheat expansin genes were obtained. To obtain more information of the expansin superfamily, genome-wide identification of the expansin genes from Brachypodium distachyon, Sorghum bicolor, Solanum lycopersicum and Gossypium raimondii genome database was performed. Based on the multiple alignments of the full-length sequences of expansins, two software’s MEGA 5.0 and MrBayes 3.2 were used to construct the phylogenetic trees, including neighbor-joining (NJ) phylogenetic tree (Han et al., 2019).
7. It need clear photographs before making any conclusive decision on this manuscript.
Response: Thanks, I have attached both PDF and TIFF files for all the figures.
Ferdous J, Hussain SS, Shi BJ. 2015. Role of microRNAs in plant drought tolerance. Plant Biotechnology Journal 13:293–305. DOI: 10.1111/pbi.12318.
Han Z, Liu Y, Deng X, Liu D, Liu Y, Hu Y, Yan Y. 2019. Genome-wide identification and expression analysis of expansin gene family in common wheat (Triticum aestivum L.). BMC Genomics 20. DOI: 10.1186/s12864-019-5455-1.
Li G, Hou M, Liu Y, Pei Y, Ye M, Zhou Y, Huang C, Zhao Y, Ma H. 2019. Genome-wide identification, characterization and expression analysis of the non-specific lipid transfer proteins in potato. BMC Genomics 20. DOI: 10.1186/s12864-019-5698-x.
Reinhart BJ, Weinstein EG, Rhoades MW, Bartel B, Bartel DP. 2002. MicroRNAs in plants. Genes and Development 16:1616–1626. DOI: 10.1101/gad.1004402.
Round 2
Reviewer 1 Report
Review of revised IJMS-543632
Novel GhNLP Genes as Identified within the QTL Region Reveal Their Role in Enhancing N-deficiency Tolerance in Cotton
Richard Odongo Magwanga, Joy Nyangasi Kirungu, Pu Lu, Xiaoyan Cai, Zhongli Zhou, Yanchao Xu, Yuqing Hou, Kunbo Wang and Fang Liu
The authors used QTL mapping by sequencing to identify 226 nodule-inception-like protein (NLP) genes in cotton (Gossypium spp.). Of these, 105 were identified in Gossypium hirsutum, 61 were identified in G. raimondii and 60 were identified in G. arboreum, respectively. They then performed bioinformatics analyses to construct a phylogeny of the NLP proteins, map their chromosomal positions, infer their molecular weights, isoelectric points and hydropathy values, and to identify miRNAs and cis-regulatory elements associated with these genes. Using RT-qPCR analysis they identified five of these genes that were significantly up-regulated under N-limited conditions. They used virus-induced gene silencing (VIGS) to knock down the expression of Gh NLP5, and showed that this compromised the ability of these plants tolerate N-limited conditions. Expression of the N-stress responsive genes, GhTap46, GhRPL18A, and GhKLU was also significantly down-regulated in these plants under N-limited conditions. These results suggest that Gh NLP5 plays an important role in tolerating N-deficiency in cotton
Overall, the analysis of the gene family provides useful information about the proteins and their evolution. The qRT-PCR and VIGS analyses also provide clues as to their potential roles in responses to N-deprivation and identify a candidate gene to use for breeding improved tolerance of N-limited conditions.
I therefore feel that this paper provides useful information worthy of publication after it has been thoroughly edited.
In their revision the authors have addressed many of my concerns. For example, the abstract now refers to the results of their phylogenetic, intorn/exon mapping and ds/dn analyses. Similarly, they now summarize the evolution of G. hirsutum in their introduction and explain the rationale for including G. raimondii and G. arboretum in their analyses. Unfortunately, many problems still persist.
Perhaps the most important is still, why do miRNAs predominantly target the D genome? Is this a mechanism to inhibit its expression, or does this mean that the D genome is the most important genome so its expression is “fine-tuned” using microRNAs? This is a very important question that must be addressed! Many crops are 4n. 6n, or more, so understanding how polyploid plants control the expression of multiple genomes enclosed within the same nucleus is a crucial problem in crop science and plant biology in general!
The English is improved, but still contains errors in nearly every sentence, starting with the title, which still contains numerous errors and still needs to explain “the QTL region.” I therefore recommend editing by a native-English speaker.
More specific comments:
1) I was unable read any of the text in figure 1, even after magnifying it to 600%, so I wasn’t able to figure out which proteins came from cotton and which were the outliers. Similarly, I couldn’t figure out the relationships between the proteins from G. hirsutum, G. raimondii and G. arboretum since I couldn’t tell which were which. Perhaps a link to a high-quality online image could be provided. In addition, the caption to figure 1 should at least explain the color-coding, and perhaps indicate the proteins from Arabdiopsis, T. cacao, and Vitis vinifera in some manner such as a different color so that readers can see where they fit into the tree.
2) The authors should be careful to indicate that the protein localizations are based on computer predictions rather than experimental validation, and they should summarize in section 2.3 the ways in which these predictions were made.
3) I am puzzled that the authors were unable to identify any cotton NLP proteins that were homologous to NLP proteins in other plants. If this is such an important family that is highly conserved, then presumably they are performing similar functions so I would expect that out of 105 NLP proteins in G. hirsutum at least one would have an ortholog in another species! If not, this needs to be discussed in the discussion! There was enough sequence conservation to identify 105 NLP proteins in G. hirsutum based on conserved domains shared with other plant species, yet they had diverged so much that not a single ortholog could be identified? This requires more attention!
4) The authors should be careful to indicate that the transcription factor and miRNA binding sites are based on computer simulations rather than experimental validation.
5) Caption to figure 4 should summarize the M&M: cotton seedlings grown hydroponically were deprived of N and then harvested at the indicated times. RNA was extracted, and the expression levels of the 53 indicated genes were analyzed by qRT-PCR using GhActin as internal control.
6) Section 2.7: the authors need to explain better why they performed this experiment and how they performed it.
7) Discussion needs to address the issues identified above.
Author Response
English language and style
(x) Extensive editing of English language and style required
Response: extensive language revision has been carried out.
|
Yes |
Can be improved |
Must be improved |
Not applicable |
|
|
Does the introduction provide sufficient background and include all relevant references? |
(x) |
( ) |
( ) |
( ) |
|
Response: Thanks |
||||
|
Is the research design appropriate? |
(x) |
( ) |
( ) |
( ) |
|
Response: Thanks |
||||
|
Are the methods adequately described? |
( ) |
(x) |
( ) |
( ) |
|
Response: The sections have been adjusted as advised |
||||
|
Are the results clearly presented? |
( ) |
(x) |
( ) |
( ) |
|
Response: adjustments done |
||||
|
Are the conclusions supported by the results? |
( ) |
(x) |
( ) |
( ) |
|
Response: Well illustrated and provides core values of the research work |
||||
Comments and Suggestions for Authors
The authors used QTL mapping by sequencing to identify 226 nodule-inception-like protein (NLP) genes in cotton (Gossypium spp.). Of these, 105 were identified in Gossypium hirsutum, 61 were identified in G. raimondii and 60 were identified in G. arboreum, respectively. They then performed bioinformatics analyses to construct a phylogeny of the NLP proteins, map their chromosomal positions, infer their molecular weights, isoelectric points and hydropathy values, and to identify miRNAs and cis-regulatory elements associated with these genes. Using RT-qPCR analysis they identified five of these genes that were significantly up-regulated under N-limited conditions. They used virus-induced gene silencing (VIGS) to knock down the expression of Gh NLP5, and showed that this compromised the ability of these plants tolerate N-limited conditions. Expression of the N-stress responsive genes, GhTap46, GhRPL18A, and GhKLU was also significantly down-regulated in these plants under N-limited conditions. These results suggest that Gh NLP5 plays an important role in tolerating N-deficiency in cotton
Response: Thanks
Overall, the analysis of the gene family provides useful information about the proteins and their evolution. The qRT-PCR and VIGS analyses also provide clues as to their potential roles in responses to N-deprivation and identify a candidate gene to use for breeding improved tolerance of N-limited conditions.
Response: Thanks
I therefore feel that this paper provides useful information worthy of publication after it has been thoroughly edited.
Response: We have carried out extensive revision on the manuscript. We do acknowledge the language error within the manuscript, we have carried out language editing, and all errors corrected.
In their revision the authors have addressed many of my concerns. For example, the abstract now refers to the results of their phylogenetic, intorn/exon mapping and ds/dn analyses. Similarly, they now summarize the evolution of G. hirsutum in their introduction and explain the rationale for including G. raimondii and G. arboretum in their analyses. Unfortunately, many problems still persist.
Perhaps the most important is still, why do miRNAs predominantly target the D genome? Is this a mechanism to inhibit its expression, or does this mean that the D genome is the most important genome so its expression is “fine-tuned” using microRNAs? This is a very important question that must be addressed! Many crops are 4n. 6n, or more, so understanding how polyploid plants control the expression of multiple genomes enclosed within the same nucleus is a crucial problem in crop science and plant biology in general!
Response: It is true that the D subgenome is very important in the evolution of tetraploid cotton, though much work is still required in elucidating the pertinent contribution of the D subgenome. However, in this research and other previous investigation, the D subgenome have been found to harbored significant QTLs and genes with higher contribution in enhancing the cotton plant survival
As explained in line 588 to 600 “The high number of miRNA target on the various genes obtained for G. raimondii, possibly could explain in part why various vital QTLs are often mapped on the D subgenome chromosomes as opposed to A subgenome chromosomes. For instance, QTL mapping of drought and salt tolerance in an introgressed recombinant inbred line population of Upland cotton, 11 QTL were detected on 8 chromosomes, in which 10 of the QTLs were located on the D subgenome [84]. Moreover, in comprehensive Meta QTL analysis for fiber quality, yield, yield-related and morphological traits, drought tolerance, and disease resistance in tetraploid cotton, A subgenome harbored 536 QTL, while the D subgenome had 687 QTL [85]. Furthermore, a detailed analysis of the RFLP map revealed that a number of resistance genes (R genes) are located on the D subgenome chromosomes compared to A subgenome chromosomes, with 5 out of 6 R genes located on the D subgenome chromosomes [86]. Moreover, analysis of the genome structure of tetraploid cotton, G. hirsutum showed that 75% (125/166) of the polymorphic loci were tagged on the D-subgenome [87]. The merger of divergent genomes in a common nucleus has been argued to present a shift from genetic flexibility to genetic fixation providing a mechanism for response to selection [88]., These findings provide a unique contribution of the D subgenome to whole-genome evolution and adaptability of tetraploid cotton to its environment, and the high miRNA target could be playing a role in fine-tuning the expression of the D subgenome genes. A number of reasons have been forwarded, explaining the significance of the D subgenome in tetraploid cotton, a higher underlying mutation rate, a higher level of DNA polymorphism, and a non-homologous chromosome rearrangement [89].
The English is improved, but still contains errors in nearly every sentence, starting with the title, which still contains numerous errors and still needs to explain “the QTL region.” I therefore recommend editing by a native-English speaker.
Response: We are very sorry for this kind of mistake, we have made corrections
More specific comments:
1) I was unable read any of the text in figure 1, even after magnifying it to 600%, so I wasn’t able to figure out which proteins came from cotton and which were the outliers. Similarly, I couldn’t figure out the relationships between the proteins from G. hirsutum, G. raimondii and G. arboretum since I couldn’t tell which were which. Perhaps a link to a high-quality online image could be provided. In addition, the caption to figure 1 should at least explain the color-coding, and perhaps indicate the proteins from Arabdiopsis, T. cacao, and Vitis vinifera in some manner such as a different color so that readers can see where they fit into the tree.
Response: Due to the huge number, the tree might not be eligible but I have provided Tiff format and the Newick structure as supplementary attachement to the manuscript.
2) The authors should be careful to indicate that the protein localizations are based on computer predictions rather than experimental validation, and they should summarize in section 2.3 the ways in which these predictions were made.
Response: Well stated
3) I am puzzled that the authors were unable to identify any cotton NLP proteins that were homologous to NLP proteins in other plants. If this is such an important family that is highly conserved, then presumably they are performing similar functions so I would expect that out of 105 NLP proteins in G. hirsutum at least one would have an ortholog in another species! If not, this needs to be discussed in the discussion! There was enough sequence conservation to identify 105 NLP proteins in G. hirsutum based on conserved domains shared with other plant species, yet they had diverged so much that not a single ortholog could be identified? This requires more attention!
Response: After careful evaluation of the phylogenetic tree, we found two G. hirsutum NLPs formed orthologous pairs as explained in line 201-209 “In analyzing the possibility of orthologous gene pair formation between the cotton NLPs and other plants used in the phylogenetic tree, only two genes were found to form orthologous gene pairs with the upland cotton NLP genes, Gh_D05G2521, and LOC_Os11g30350 (NLP gene from Oryza sativa); and the other pair was, Ghsca101252G01 and GSVIVG01026649001 (NLP gene from Vitis vinifera). None of the orthologous gene pairs was formed between cotton and the NLP proteins obtained for T. cacao the closest relative of the genus Gossypium compared to other plant species analyzed.”
4) The authors should be careful to indicate that the transcription factor and miRNA binding sites are based on computer simulations rather than experimental validation.
Response: well stated
5) Caption to figure 4 should summarize the M&M: cotton seedlings grown hydroponically were deprived of N and then harvested at the indicated times. RNA was extracted, and the expression levels of the 53 indicated genes were analyzed by qRT-PCR using GhActin as internal control.
Response: Corrected
6) Section 2.7: the authors need to explain better why they performed this experiment and how they performed it.
Response: Well stated
7) Discussion needs to address the issues identified above.
Response: we have factored in all as advised
Round 3
Reviewer 1 Report
Review of ijms-543632 version 3
In their revision the authors have addressed many of my concerns. For example, they have now described how the subcellular localizations were performed in section 2.3, they have addressed the issue of NLP orthologs in other plant species and they have improved the caption to figure 4. They have also addressed why more miRNA target the D genome than the A-genome.
Unfortunately, many problems still persist.
1) At the moment the paper is very descriptive, and I would like to see more analysis. Why are over half of the NLP proteins targeted to the nucleus? Are they transcription factors, or do they perform some other function in nitrogen signaling? What about the proteins targeted to the ER or the plasma membrane? What might the proteins localized in the chloroplasts be doing? I realize that they don’t have hard data, but since they speculate on the significance of the proteins being hydrophilic they can think a bit more about what else they might be doing.
2) In the abstract the authors refer to profiling under drought-stress, yet they do not provide any drought-stress data in this paper. They need to clarify that their previous study of drought stress identified a gene which they thought might also be involved in responding to N-deficiency. They might also want to consider removing this sentence altogether.
There also seems to be some confusion about the motivation for studying NLP proteins, for example on lines 199 and 610. Was this study performed because of the potential importance of NLP proteins or because an NLP was identified in the screen for drought tolerance?
3) miRNA are usually 21 nt long
4) There is still confusion between computational predictions and experimental validation, for example, in identifying miRNA binding sites. The authors identified multiple potential miRNA targets, but none were experimentally validated.
5) The caption for figure 4 contains numerous English errors that must be corrected
6) Section 2.7 is repetitive and needs to be more concise
The English is improved, but still contains numerous errors. I have corrected the abstract to illustrate just how much work is needed, but I don’t have time to correct the entire paper.
Line 3 should be: “…Reveals their Roles in…”
Line 18: ”… boost its…” should be “…boost their…”
Line 19: “approach” should be “approaches”
Line 20: “occasioned by” should be “due to”
Line 21 should be “…genotypes, an abiotic…”
Lines 22-23 should be “…and a highly-susceptible, but very productive, G. hirsutum…”
Lines 23-25: These must be rewritten to clarify that this is previous work reported elsewhere. You did not report any profiling under drought stress in this study! Instead, you tested the role of the gene identified in the previous study in coping with N-deficiency! Alternatively, consider removing this sentence altogether.
Line 23 should be “drought stress conditions.”
Line 24: “sequences” should be “sequencing.”
Line 25 should be “…integrated to map drought-tolerant…”
Line 25: the authors must explain “the stable QTLs region…” Is this the QTL identified associated with drought tolerance in your previous study?
Line 26 should be “… region, a nodule-inception-like protein (NLP) gene was identified. We…”
Line 28 should be “…most NLP genes…”
Lines 29-30 should be “Moreover, functions of one of the highly upregulated genes, Gh_A05G3286 (Gh NLP5), were evaluated using the virus gene silencing (VIGS) mechanism “
Line 32 and line 954 should be “Comprehensive in silico analysis…”
Lines 33-35 should be “…had varying molecular weights, protein lengths, isoelectric points (pI), and grand hydropathy values (GRAVY). These ranged from negative one to zero, showing that the proteins were hydrophilic.
Lines 35-37 should be: “…Moreover, various cis-regulatory elements that are the binding sites for stress-associated transcription factors were found in the promoters of various NLP genes. In addition, many miRNAs were predicted to target NLP genes, notably miR167a, miR167b, miR160, and miR167 which were previously shown to
to target five NAC genes, including NAC1 and CUC1, under N-limited conditions.
Lines 38-39: Are you reporting the results of the previous study? You did not report any over-expression data in this paper!
Line 42 should be “…regulating plant responses to N-limited …”
Line 43 should be “…the knockdown of the Gh_A05G3286 (GhNLP5) gene by virus-induced silencing (VIGS) significantly reduced the ability of these plants to…”
Lines 46-72 should be “…addition to higher levels of malondialdehyde (MDA) and significantly reduced levels of proline and superoxide dismutase (SOD) compared to the WT under N-limited conditions. Subsequently, the expression levels of the nitrogen-stress responsive genes, GhTap46, GhRPL18A, and GhKLU were shown to be significantly…”
Other specific comments
Lines 330, 338, 854 : Is it wolfsport or wolfpsort?
Line 399: Explain that you are looking for potential binding sites for stress-associated transcription factors as an indication that they encode proteins involved in stress responses.
Line 405: rephrase to indicate that did computer search for transcription factor binding sites. These elements are just the binding sites for TFs and do not function as elicitors and are not induced or are not responsive.
Line 410: detection of these elements only shows that they are potentially critical in enhancing survival under abiotic stress conditions.
Lines 421-422: rephrase to state that these are computer-predicted targets and are not experimentally-validated.
Lines 431-434 are confusing and must be rewritten for clarity.
Line 442: Start by explaining that the purpose of this experiment was to identify NLP genes that were significantly up-regulated under N-deficiency and might therefore be candidates for regulating N-deficiency responses.
Line 500: please explain what the TRV:PDS is and its purpose before describing the results. The way you do it on lines 898-899 would be fine.
Line 518: Please explain that the TRV:00 is the empty-vector control.
Line 566: proline is not an enzyme! Do you mean two markers of oxidative stress? Also, please explain what the MDA concentration tells you.
Lines 575-580 should be broken into at least two sentences.
Lines 636-640 should be broken into at least two sentences.
Lines 640-651 should be broken into at least two sentences and rewritten for clarity. Also, anyone reading this paper will know the basics of transcription and translation!
Lines 663-665: please comment on why it is is significant that most NLP proteins localize to the nucleus. Are they transcription factors? If not, what other role might they serve?
Lines 665-671 seem unnecessary as written. Is there evidence that the NLP genes were originally on the plastid genome? Is there evidence that NLP proteins play a role in nitrogen assimilation in the plastids? Please explain better or delete this section.
Lines 679-683 should be broken into at least two sentences. You also need to explain better how the yeast data shines light on the potential role of NLP proteins in plants. If they work in the nucleus, why are there so many in the ER and in the plasma membrane?
Line 689: reiterate that this a prediction without experimental support.
Lines 710-713 should be broken into two sentences.
Line 714: miR164 was predicted to target four genes
Lines 761- 774 should be broken into at least two sentences.
Lines 795-801 and 906 – 913 are essentially identical. Rewrite lines 906-913 to state that N-deficiency was created as described above in section 4.1, Plant material s and treatments.
Lines 868-871 should be broken into at least two sentences.
Lines 878 – 880 should be broken into two sentences.
Author Response
Review of ijms-543632 version 3
In their revision the authors have addressed many of my concerns. For example, they have now described how the subcellular localizations were performed in section 2.3, they have addressed the issue of NLP orthologs in other plant species and they have improved the caption to figure 4. They have also addressed why more miRNA target the D genome than the A-genome.
Unfortunately, many problems still persist.
1) At the moment the paper is very descriptive, and I would like to see more analysis. Why are over half of the NLP proteins targeted to the nucleus? Are they transcription factors, or do they perform some other function in nitrogen signaling? What about the proteins targeted to the ER or the plasma membrane? What might the proteins localized in the chloroplasts be doing? I realize that they don’t have hard data, but since they speculate on the significance of the proteins being hydrophilic they can think a bit more about what else they might be doing.
Response: the main aim of this research work was to unravel the possible role of the NLP proteins, which we have done through a credible method, the RNAi technique which has been extensively used by a number of researchers globaly in determining the functions of various plants functional genes. we went further to carry out intensive bioinformatics to provide further evidence of the possible involvement of the NLP proteins in enhancing tolerwnce levels to N defiency.
2) In the abstract the authors refer to profiling under drought-stress, yet they do not provide any drought-stress data in this paper. They need to clarify that their previous study of drought stress identified a gene which they thought might also be involved in responding to N-deficiency. They might also want to consider removing this sentence altogether.
Response: The research is a build up from our previous work, all the correction done as advised.
There also seems to be some confusion about the motivation for studying NLP proteins, for example on lines 199 and 610. Was this study performed because of the potential importance of NLP proteins or because an NLP was identified in the screen for drought tolerance?
Response: This research was done due to the potential role of the NLP proteins in enhancing N deficiency tolerance.
3) miRNA are usually 21 nt long
Response: changed
4) There is still confusion between computational predictions and experimental validation, for example, in identifying miRNA binding sites. The authors identified multiple potential miRNA targets, but none were experimentally validated.
Response: We predicted the miRNA target in order to compare with other researchers so far done in relation to the NLPs, we identified some critical miRNAs whih have been found to target critical genes, thus this was done to provide further evidence that the NLP proteins could perharps be critical for the plants under N-limited condition
5) The caption for figure 4 contains numerous English errors that must be corrected
Response: corrected
6) Section 2.7 is repetitive and needs to be more concise
Response: changed
The English is improved, but still contains numerous errors. I have corrected the abstract to illustrate just how much work is needed, but I don’t have time to correct the entire paper.
Line 3 should be: “…Reveals their Roles in…”
Response: changed
Line 18: ”… boost its…” should be “…boost their…”
Response: changed
Line 19: “approach” should be “approaches”
Response: changed
Line 20: “occasioned by” should be “due to”
Response: changed
Line 21 should be “…genotypes, an abiotic…”
Response: changed
Lines 22-23 should be “…and a highly-susceptible, but very productive, G. hirsutum…”
Response: changed
Lines 23-25: These must be rewritten to clarify that this is previous work reported elsewhere. You did not report any profiling under drought stress in this study! Instead, you tested the role of the gene identified in the previous study in coping with N-deficiency! Alternatively, consider removing this sentence altogether.
Response: corrected
Line 23 should be “drought stress conditions.”
Response: changed
Line 24: “sequences” should be “sequencing.”
Response: changed
Line 25 should be “…integrated to map drought-tolerant…”
Response: changed
Line 25: the authors must explain “the stable QTLs region…” Is this the QTL identified associated with drought tolerance in your previous study?
Response: Added
Line 26 should be “… region, a nodule-inception-like protein (NLP) gene was identified. We…”
Response: changed
Line 28 should be “…most NLP genes…”
Response: changed
Lines 29-30 should be “Moreover, functions of one of the highly upregulated genes, Gh_A05G3286 (Gh NLP5), were evaluated using the virus gene silencing (VIGS) mechanism “
Response: changed
Line 32 and line 954 should be “Comprehensive in silico analysis…”
Response: changed
Lines 33-35 should be “…had varying molecular weights, protein lengths, isoelectric points (pI), and grand hydropathy values (GRAVY). These ranged from negative one to zero, showing that the proteins were hydrophilic.
Response: changed
Lines 35-37 should be: “…Moreover, various cis-regulatory elements that are the binding sites for stress-associated transcription factors were found in the promoters of various NLP genes. In addition, many miRNAs were predicted to target NLP genes, notably miR167a, miR167b, miR160, and miR167 which were previously shown to
to target five NAC genes, including NAC1 and CUC1, under N-limited conditions.
Response: changed
Lines 38-39: Are you reporting the results of the previous study? You did not report any over-expression data in this paper!
Response: changed
Line 42 should be “…regulating plant responses to N-limited …”
Response: changed
Line 43 should be “…the knockdown of the Gh_A05G3286 (GhNLP5) gene by virus-induced silencing (VIGS) significantly reduced the ability of these plants to…”
Response: changed
Lines 46-72 should be “…addition to higher levels of malondialdehyde (MDA) and significantly reduced levels of proline and superoxide dismutase (SOD) compared to the WT under N-limited conditions. Subsequently, the expression levels of the nitrogen-stress responsive genes, GhTap46, GhRPL18A, and GhKLU were shown to be significantly…
Response: changed”
Other specific comments
Lines 330, 338, 854 : Is it wolfsport or wolfpsort?
Response: WoLF PSORT
Line 399: Explain that you are looking for potential binding sites for stress-associated transcription factors as an indication that they encode proteins involved in stress responses.
Response: corrected
Line 405: rephrase to indicate that did computer search for transcription factor binding sites. These elements are just the binding sites for TFs and do not function as elicitors and are not induced or are not responsive.
Response: changed
Line 410: detection of these elements only shows that they are potentially critical in enhancing survival under abiotic stress conditions.
Lines 421-422: rephrase to state that these are computer-predicted targets and are not experimentally-validated.
Response: changed
Lines 431-434 are confusing and must be rewritten for clarity.
Response: changed
Line 442: Start by explaining that the purpose of this experiment was to identify NLP genes that were significantly up-regulated under N-deficiency and might therefore be candidates for regulating N-deficiency responses.
Response: changed
Line 500: please explain what the TRV:PDS is and its purpose before describing the results. The way you do it on lines 898-899 would be fine.
Response: explained
Line 518: Please explain that the TRV:00 is the empty-vector control.
Response: done
Line 566: proline is not an enzyme! Do you mean two markers of oxidative stress? Also, please explain what the MDA concentration tells you.
Response: corrected and information added
Lines 575-580 should be broken into at least two sentences
Response: done.
Lines 636-640 should be broken into at least two sentences.
Response: changed
Lines 640-651 should be broken into at least two sentences and rewritten for clarity. Also, anyone reading this paper will know the basics of transcription and translation!
Response: changed
Lines 663-665: please comment on why it is is significant that most NLP proteins localize to the nucleus. Are they transcription factors? If not, what other role might they serve?
Response: the NLPs are genes and the presence of the proteins encoded by these genes could explain their vital role in enhancing plants response to N-limited contions.
Lines 665-671 seem unnecessary as written. Is there evidence that the NLP genes were originally on the plastid genome? Is there evidence that NLP proteins play a role in nitrogen assimilation in the plastids? Please explain better or delete this section.
Response: the is a further explanation that the high number of the NLP proteins within the nucleus could be justified, and their high number could be attributed to duplications.
Lines 679-683 should be broken into at least two sentences. You also need to explain better how the yeast data shines light on the potential role of NLP proteins in plants. If they work in the nucleus, why are there so many in the ER and in the plasma membrane?
Response: no yeast experiment was carried out, howevr the correction is done as advised
Line 689: reiterate that this a prediction without experimental support.
Response: done
Lines 710-713 should be broken into two sentences.
Response: done
Line 714: miR164 was predicted to target four genes
Response: changed
Lines 761- 774 should be broken into at least two sentences.
Response: done
Lines 795-801 and 906 – 913 are essentially identical. Rewrite lines 906-913 to state that N-deficiency was created as described above in section 4.1, Plant material s and treatments.
Response: changed
Lines 868-871 should be broken into at least two sentences.
Response: changed
Lines 878 – 880 should be broken into two sentences.
Response: changed
Round 4
Reviewer 1 Report
Review of v4 IJMS-543632
Map-Based Functional Analysis of the GhNLP Genes Reveals Their Roles in Enhancing Tolerance to N-Deficiency in Cotton
Richard Odongo Magwanga, Joy Nyangasi Kirungu, Pu Lu, Xiaoyan Cai, Zhongli Zhou, Yanchao Xu, Yuqing Hou, Kunbo Wang and Fang Liu
The authors used QTL mapping by sequencing to identify 226 nodule-inception-like protein (NLP) genes in cotton (Gossypium spp.). Of these, 105 were identified in Gossypium hirsutum, 61 were identified in G. raimondii and 60 were identified in G. arboreum, respectively. They then performed bioinformatics analyses to construct a phylogeny of the NLP proteins, map their chromosomal positions, infer their molecular weights, isoelectric points and hydropathy values, and to identify miRNAs and cis-regulatory elements associated with these genes. Using RT-qPCR analysis they identified five of these genes that were significantly up-regulated under N-limited conditions. They used virus-induced gene silencing (VIGS) to knock down the expression of Gh NLP5, and showed that this compromised the ability of these plants tolerate N-limited conditions. Expression of the N-stress responsive genes, GhTap46, GhRPL18A, and GhKLU was also significantly down-regulated in these plants under N-limited conditions. These results suggest that Gh NLP5 plays an important role in tolerating N-deficiency in cotton
Overall, the analysis of the gene family provides useful information about the proteins and their evolution. The qRT-PCR and VIGS analyses also provide clues as to their potential roles in responses to N-deprivation and identify a candidate gene to use for breeding improved tolerance of N-limited conditions.
In this revision the authors have addressed many of my concerns, except that I would like them to explain better why miRNA almost exclusively target genes from the D genome.
However, there are still many problems with the English. Here are some examples, but there are many more that need correction
Line 87: Are NIN proteins the same as NLP proteins?
Lines 88-89 should be “Even in non-leguminous plants, they function as nodule initiators and regulate the number of nodules that are formed.”
Lines 91-93 should be “Moreover, overexpression of two maize NODULE-INCEPTION-like proteins, the ZmNLP6 and ZmNLP8, in the Arabidopsis, restored the nitrate assimilation and induction of the nitrate-responsive genes in the Arabidopsis NLP7-4 mutant. This restoration of
Line 105 should be “…functions, acting upstream …”
Line 112 should be “…VIGS-plants…”
Line 186 should be “…ds/dn value greater than 1…”
Author Response
Open Review
(x) I would not like to sign my review report
( ) I would like to sign my review report
English language and style
( ) Extensive editing of English language and style required
( ) Moderate English changes required
(x) English language and style are fine/minor spell check required
Response: Thanks so much, but I has effected all the changes.
( ) I don't feel qualified to judge about the English language and style
|
Yes |
Can be improved |
Must be improved |
Not applicable |
|
|
Does the introduction provide sufficient background and include all relevant references? |
(x) |
( ) |
( ) |
( ) |
|
Response: Thanks |
||||
|
Is the research design appropriate? |
(x) |
( ) |
( ) |
( ) |
|
Response: Thanks |
||||
|
Are the methods adequately described? |
(x) |
( ) |
( ) |
( ) |
|
Response: Thanks |
||||
|
Are the results clearly presented? |
( ) |
(x) |
( ) |
( ) |
|
Response: I have made adjustments, and i factored in the miRNA |
||||
|
Are the conclusions supported by the results? |
( ) |
(x) |
( ) |
( ) |
|
Response: Thanks, I have effected the changes and integrated all the major findings as obtained. |
||||
Comments and Suggestions for Authors
Review of v4 IJMS-543632
Map-Based Functional Analysis of the GhNLP Genes Reveals Their Roles in Enhancing Tolerance to N-Deficiency in Cotton
Richard Odongo Magwanga, Joy Nyangasi Kirungu, Pu Lu, Xiaoyan Cai, Zhongli Zhou, Yanchao Xu, Yuqing Hou, Kunbo Wang and Fang Liu
The authors used QTL mapping by sequencing to identify 226 nodule-inception-like protein (NLP) genes in cotton (Gossypium spp.). Of these, 105 were identified in Gossypium hirsutum, 61 were identified in G. raimondii and 60 were identified in G. arboreum, respectively. They then performed bioinformatics analyses to construct a phylogeny of the NLP proteins, map their chromosomal positions, infer their molecular weights, isoelectric points and hydropathy values, and to identify miRNAs and cis-regulatory elements associated with these genes. Using RT-qPCR analysis they identified five of these genes that were significantly up-regulated under N-limited conditions. They used virus-induced gene silencing (VIGS) to knock down the expression of Gh NLP5, and showed that this compromised the ability of these plants tolerate N-limited conditions. Expression of the N-stress responsive genes, GhTap46, GhRPL18A, and GhKLU was also significantly down-regulated in these plants under N-limited conditions. These results suggest that Gh NLP5 plays an important role in tolerating N-deficiency in cotton
Response: Thanks so much, but I have effected all the changes.
Overall, the analysis of the gene family provides useful information about the proteins and their evolution. The qRT-PCR and VIGS analyses also provide clues as to their potential roles in responses to N-deprivation and identify a candidate gene to use for breeding improved tolerance of N-limited conditions.
Response: Thanks so much, but I have effected all the changes.
In this revision the authors have addressed many of my concerns, except that I would like them to explain better why miRNA almost exclusively target genes from the D genome.
Response: The high number of miRNA target on the various genes obtained for G. raimondii, possibly could explain in part why various vital QTLs are often mapped on the D subgenome chromosomes as opposed to A subgenome chromosomes. Polyploidization contributed significantly to the alteration of the gene functions in the tetraploid cotton, moreover, higher number of transcription factors in tetraploid cotton are contributed by Dt subgenomes compared to the At subgenome, furthermore, previous reports have shown that a number of significant QTLs were mapped in Dt subgenome than At subgenome [87]. For instance, QTL mapping of drought and salt tolerance in an introgressed recombinant inbred line population of Upland cotton, 11 QTL were detected on 8 chromosomes, in which 10 of the QTLs were located on the D subgenome [88]. Moreover, in comprehensive Meta QTL analysis for fiber quality, yield, yield-related and morphological traits, drought tolerance, and disease resistance in tetraploid cotton, A subgenome harbored 536 QTL, while the D subgenome had 687 QTL [89].
However, there are still many problems with the English. Here are some examples, but there are many more that need correction
Line 87: Are NIN proteins the same as NLP proteins?
Response: changed.
Lines 88-89 should be “Even in non-leguminous plants, they function as nodule initiators and regulate the number of nodules that are formed.”
Response: changed
Lines 91-93 should be “Moreover, overexpression of two maize NODULE-INCEPTION-like proteins, the ZmNLP6 and ZmNLP8, in the Arabidopsis, restored the nitrate assimilation and induction of the nitrate-responsive genes in the Arabidopsis NLP7-4 mutant. This restoration of
Response: changed
Line 105 should be “…functions, acting upstream …”
Response: changed
Line 112 should be “…VIGS-plants…”
Response: changed
Line 186 should be “…ds/dn value greater than 1…”
Response: corrected